# A bioinformatics approach to design minimal biomimetic metal-binding peptides
Claudia Spallacci [1], Marco Chino [2], Antonio Rosato[3,4], Ornella Maglio [2,5], Ping Huang [1], Luca D'Amario [1], Angela Lombardi [2] ✉, Claudia Andreini[3,4] ✉ & Mun Hon Cheah [1] ✉

Nature-inspired or biomimetic catalysts aim to reach the high catalytic performance and selectivity of natural enzymes while possessing the chemical stability and processability of synthetic catalysts. A promising strategy for designing biomimetic catalysts relies on mimicking the structure of the enzyme active site. This can either entail complicated total synthesis of a synthetic catalyst or design of peptide sequences, able to self-assemble in the presence of metal ions, thus forming metallo-peptide complexes that mimic the active sites of natural enzymes. Using a bioinformatics approach, we designed a minimal peptide made up of eight amino acids (H4pep) to act as a functional mimic of the trinuclear Cu site of the laccase enzyme. Cu(II) binding to H4pep results in the formation of a $Cu^{2+}(H4pep)_2$ complex with a β-sheet secondary structure, able to reduce $O_2$. Our study demonstrates the viability and potential of using short peptides to mimic the minimal functional site of natural enzymes.

The growing demand for sustainable energy, coupled with the climate crisis, motivates the development of efficient devices for storing and producing renewable fuels and chemicals. Nature has taken advantage of billions of years of evolution to carefully optimize enzyme structures for catalyzing specific chemical reactions. Nature's repertoire of enzyme functions is coupled with the fundamental ability to accept a variety of different substrates, a promiscuity that is at the base of evolution[1]. Metalloenzymes offer a rich library of models for studying and developing artificial catalysts able to drive reactions crucial to the renewable energy transition[2,3]. The Protein Data Bank (PDB) is the leading global repository for proteins, nucleic acids, carbohydrates, and other experimentally determined biological complexes[4]. Recent advancements in metalloenzymes research have shown that the residues in close proximity to the metal-binding site are the main determinants in driving catalytic reaction bias and selectivity[5]. In this context, mini-proteins and biomimetic peptide design, with their vastly simplified structure, have emerged as valuable tools to examine structure-function relationships influencing metal selectivity and reactivity[6–9]. The appeal of using short peptides as scaffolds to build efficient catalysts lies in their versatility and potential to achieve high catalytic activity rivaling natural enzymes while possessing higher thermal and pH stability[10,11]. Additionally, minimal peptides are more amenable towards interfacial electron transfer on electrodes due to their smaller cross-section, thereby achieving higher surface coverage on electrode surfaces. Moreover, short peptides can be easily synthesized in large amounts and at low cost, thanks to the advanced techniques and well-established methods available in solid-phase peptide synthesis (SPPS)[12,13].

With this in mind, we have chosen to use natural metalloenzymes as models to design short peptide ligands as a scaffold for supporting the self-assembly of biomimetic catalysts. Focusing on the first coordination sphere around the catalytic cofactor of target metalloenzymes, our goal is to identify the shortest peptide sequence able to bind metals and mimic the model's site activity. In this regard, the concept of "minimal functional site" (MFS) developed by Andreini et al. is a good starting point for the design, as it describes the minimal environment determining the metal's chemical behavior[14]. MFSs represent the local three-dimensional environment, including all residues within 5 Å distance from any metal-binding ligand, providing a tool for classifying and comparing enzymatic metal sites. Bioinformatic tools can easily extract MFSs information, enabling their manipulation and supporting the design of new biomimetic molecules. The potential of bioinformatic approaches for designing bioactive peptides and molecules is well-established, with examples in drug design, vaccine development, and catalysis[15–18].

[1]Molecular Biomimetics, Department of Chemistry-Ångström Laboratory, Uppsala University, Uppsala, Sweden. [2]Department of Chemical Sciences, University of Naples Federico II, Napoli, Italy. [3]Department of Chemistry, University of Florence, Sesto Fiorentino, Italy. [4]Magnetic Resonance Center (CERM), University of Florence, Sesto Fiorentino, Italy. [5]Institute of Biostructures and Bioimaging, National Research Council, Napoli, Italy. ✉e-mail: alombard@unina.it; claudia.andreini@unifi.it; michael.cheah@kemi.uu.se

We have thus created a bioinformatic tool, MetalSite-Analyzer (MeSA, https://metalsite-analyzer.cerm.unifi.it/), that enables users to extract relevant sequence motifs for binding a metal of choice. The tool leverages MFS sequence alignments to obtain information on the most conserved residues in metal sites belonging to the protein family of interest. To demonstrate the value of this approach, we have designed a minimal eight-residue peptide (H4Pep), using the trinuclear copper site of laccase as a model. This well-known enzyme was chosen given our interest in redox catalysis. The design of the minimal peptide was based both on sequence conservation analysis, as dictated by MeSA, and on structural modification, led by rational observation of the enzyme's three-dimensional structure.

The ability of H4pep to coordinate copper ions was verified via different spectroscopic methods. Notably, the CD data demonstrate that H4pep-$Cu^{2+}$ complexes adopt a beta-sheet conformation, as it is indeed observed in the MFS of laccase. Our data demonstrate that, despite their beta-sheet conformation, H4pep-$Cu^{2+}$ complexes do not undergo aggregation or formation of fibrils. We also performed preliminary catalytic activity measurements to ascertain if these metallo-peptide complexes exhibit activity based on their parent laccase. Our assessment focuses on indications of positive catalytic activity with respect to $O_2$ reduction, serving as proof-of-concept for the validity of the proposed MeSA tool. To the best of our knowledge, this is the first example of a synthetic β-sheet metallo-peptide complex that is stable in solution and features catalytic activity. These findings demonstrate that the MetalSite-Analyzer tool can provide relevant indications for designing bioinspired catalytically active metallo-peptides.

## Results
### Bioinformatic tool and peptide design
We have created the MeSA tool with the scope of expanding the MFS concept to include sequence conservation analysis. Starting from an input PDB structure, MeSA allows the selection of user-defined mono-/multi-nuclear metal sites and, from the extracted MFS, runs a PSI-BLAST search of the metal-binding sequence fragments[19]. This step consists of mapping and aligning the binding fragments to all related sequences contained in Uniprot, the world-leading protein sequence database[20]. The output information allows for the analysis of the conservation of the residues in each specific position of the starting MFS sequence. This provides a basis for further rational design of the desired peptide mimics, by highlighting completely conserved residues, which are presumably strictly necessary for function, moderately variable positions, where one of two/three different amino acids can be selected (e.g., based on considerations of stability or ease of synthesis), and highly variable positions, where almost any amino acid can be introduced. The tool has been implemented as a user-friendly web server, requiring no registration (https://metalsite-analyzer.cerm.unifi.it/).

To assess our methodological approach, we have selected laccase as a model to design a minimal-length peptide ligand for binding copper ions. Laccases belong to the family of multicopper oxidases, which catalyze one-electron oxidation of different organic substrates, coupling it with the four-electron reduction of $O_2$ to $H_2O$[21]. In these enzymes, substrate oxidation occurs at a mononuclear copper site (type 1), followed by electron transfer to a trinuclear copper cluster (type 2/type 3), where oxygen reduction occurs. The latter is the site we have selected to test our bioinformatic tool.

Firstly, we accessed the MetalPDB database[22,23] to gain information on the binding site(s) of the small laccase from *Streptomyces viridosporus* (PDB ID: 3tbc). In bacterial small laccases, so-called because they have fewer domains than fungal laccases, the trinuclear Cu sites are located at the interface between monomers, in a three-fold symmetry. As in a typical multicopper oxidase, they consist of eight histidine residues coordinating the metals, in a ligand-nonligand-ligand motif (LXL)[24,25]. In the apo-form of the enzyme, the monomer exposes the coordinating residues to the surrounding environment, making it potentially accessible to solvent and substrate molecules involved in catalysis. This makes the site particularly attractive to inspire the design of artificial biomimetic peptide catalysts. Feeding the pdb structure as input to MeSA and selecting the trinuclear

copper site, the tool was able to extract four fragments contributing to the coordination of the Cu ions with two histidine residues each (Fig. 1A). The conservation of the residues in each fragment was analyzed via the PSI-BLAST algorithm. As a result, four alignments of metal-binding motifs were generated. Finally, a sequence profile[26] was obtained for each fragment composing the starting site (Fig. 1B).

In our endeavor to obtain a minimal-length peptide mimicking the laccase site's activity, further rational design steps are needed to implement the information obtained from MeSA. On a closer look at the structure of the site, we have noticed that the binding His residues belong two by two to antiparallel β-sheets, located on adjacent monomers in a C2 symmetry with respect to the axis running across the Cu atoms (Fig. 1C). The four metal-binding fragments identified by our bioinformatic search thus represent the four β-strands composing this structural motif. In particular, two of the four fragments extracted by the bioinformatic search are considerably shorter (7–9 residues for fragment 1 of chain A and B versus 11–13 for fragment 2 of chain A and B, see Fig. 1B), thus more suited for our purpose of designing a minimal-length peptide. Hence, for the final sequence, the choice of the amino acid residues in each position was driven by the combination of the consensus sequences of these two fragments, with further modifications led by the rational analysis of the model site (see "Methods" for details). The final minimal-length peptide ligand is eight residues long and was named H4pep (sequence: HTVHYHGH).

### Binding of metal ions by H4pep
H4pep was synthesized via SPPS and purified via reverse-phase high-pressure liquid chromatography (RP-HPLC). To verify the consistency of our design strategy, we preliminarily carried out UV-visible and NMR experiments to confirm the ability of H4pep to bind copper ions. Such experiments were performed at pH 5.6 to prevent copper oxide formation and N-terminal amine deprotonation (*vide infra*), still favoring imidazole deprotonation and binding (pKa ~6)[27].

UV-visible spectra of H4pep display the characteristic peak of tyrosine at 276 nm (Fig. 2A). Upon gradual copper(II) addition, the characteristic band of $Cu^{2+}$ d-d transitions arises in the region between 500 and 800 nm. This feature is in good agreement with reported $\lambda_{max}$ values for histidine coordination in peptide-copper complexes[28]. Notably, with copper concentration up to 1 equivalent, this band could be fitted with two Gaussian functions, suggesting the contribution of multiple species to the absorbance profile (see "Stoichiometry of H4pep-$Cu^{2+}$ complexes" section for details). In excess of copper, the absorption peak of unbound copper(II), with maxima at 786 nm, becomes noticeable and increasingly contributes to the observed spectrum.

Binding of copper(I) was also tested to verify the ability of H4pep to bind copper also in the reduced oxidation state. The affinity of H4pep to $Cu^+$ ions was determined via competitive titration with bicinchoninic acid (BCA, Fig. S1). The absorption changes at 562 nm, corresponding to the formation of $[Cu(BCA)_2]^{3-}$, could be fitted satisfactorily assuming a single class of binding sites with $K_D = 2.4*10^{-12}$ M. This affinity is comparable to the values found for a series of nitrite reductase mimics featuring a Cu(His)$_3$ site bound to a triple-stranded α-helical coiled-coil[29,30].

To further investigate the coordination of metal ions to H4pep, the peptide's structure was analyzed by NMR spectroscopy under different experimental conditions. In aqueous solution, the apo-peptide showed only four very broad amide signals of the nine expected (seven backbone and two C-terminal amide protons) due to fast exchange with the solvent, indicating the absence of stable hydrogen bonds and secondary structure elements (Figs. S2A and S3A). In methanol, while amide resonances became observable, their chemical shift dispersion remained limited, suggesting lack of defined tertiary structure in the apo-form (Figs. S3B and S4A). The use of $Zn^{2+}$ as a diamagnetic probe in NMR studies is a strategy used to gain preliminary structural insights when the paramagnetic nature of $Cu^{2+}$ precludes direct NMR analysis[31], including in the particularly demanding pentacoordinated geometry found in LPMOs[32,33]. In such cases, $Zn^{2+}$ allows for the identification of potential binding sites and ligand interactions without the complications associated with paramagnetic broadening. The

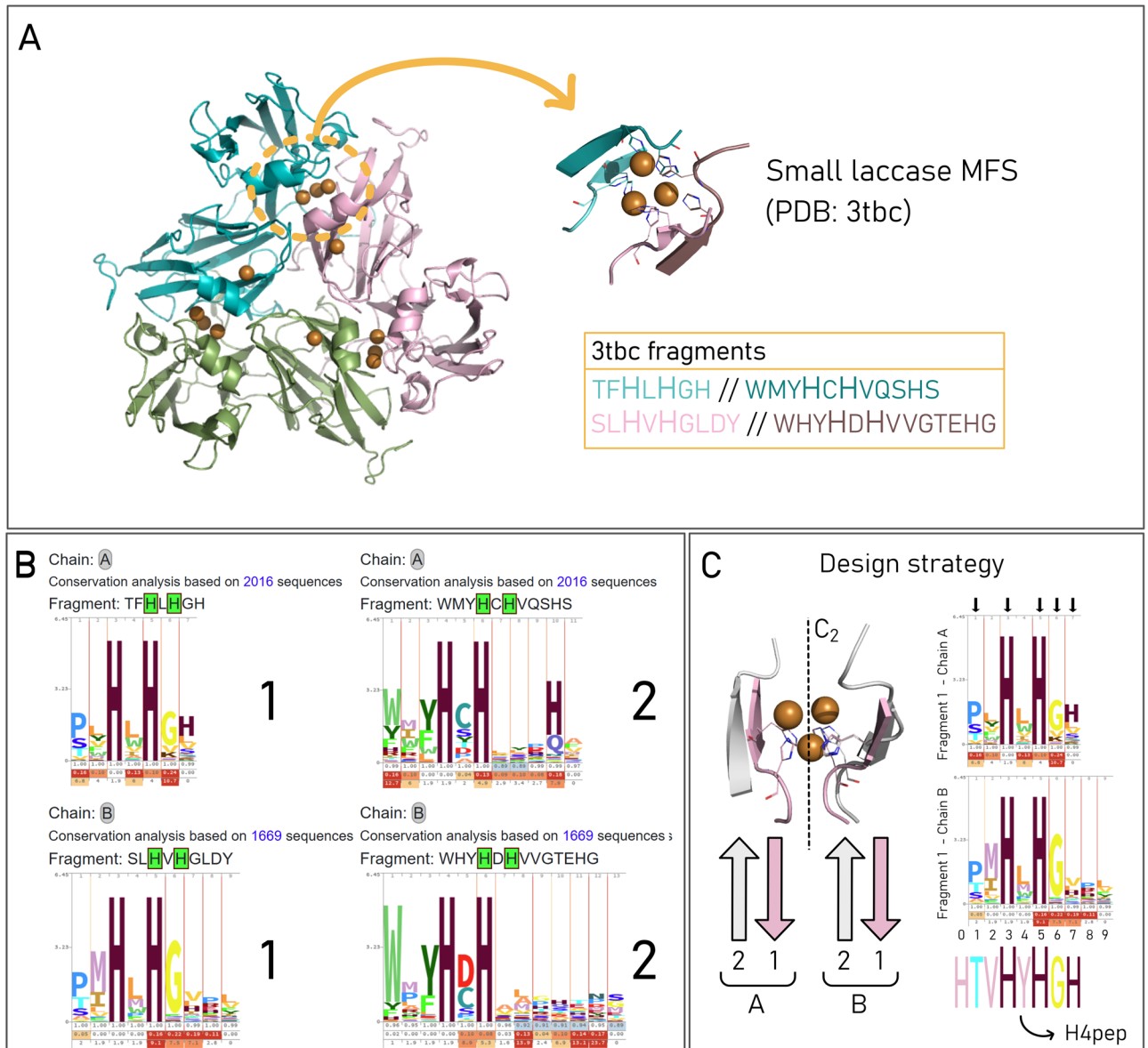

**Fig. 1 | Model laccase, MetalSite-Analyzer output, and design strategy. A** PDB structure of the small laccase used as a model for our design (PDB ID: 3tbc) with focus on its minimal functional site (MFS) and binding fragments extracted by MetalSite-Analyzer (MeSA). **B** Output of MeSA based on the trinuclear copper site of the model small laccase. 1 and 2 represent fragments belonging to chain A (top) and to chain B (bottom). **C** The model MFS features a $C_2$ symmetry axis running across the Cu atoms; the binding histidines of the shortest fragments (1 from chain A and 1 from chain B) show a high degree of overlap according to this symmetry. Thus, these fragments are selected for the following sequence variability analysis. For clarity, the binding histidines belonging to other fragments are not shown. The selection of the final sequence is represented graphically on the right. Conserved residues, including coordinating His, are highlighted using black arrows (see "Methods" for details).

addition of $Zn^{2+}$ in a 1:1 ratio induced selective changes in the histidine resonances, with downfield shifts (~0.1 ppm) of the proton signals belonging to the Cδ and Cε of the imidazole ring (Fig. S2B). The broadening of these resonances suggests a chemical exchange process, widely observed in NMR analysis, which primarily involves the histidine residues[34]. This spectral behavior suggests a dynamic equilibrium between multiple zinc-bound species, where either one or two $Zn^{2+}$ ions are alternately coordinated to the two different available binding sites in the peptide scaffold. More dramatic spectral changes were observed upon addition of $Cu^+$ (at H4pep:$Cu^+$ ratios of 1:1 and 2:1) under anaerobic conditions, both in aqueous buffer (Figs. S2C and S5) and methanol (Figs. S4B and S6). All resonances significantly broadened, suggesting that $Cu^+$ binding affects the overall peptide conformation more extensively than $Zn^{2+}$, leading to multiple conformational states in exchange. The spectral broadening persisted

in the presence of sodium dithionite, excluding contributions from paramagnetic $Cu^{2+}$ species. From the above results, we can conclude that H4pep binds both $Zn^{2+}$ and $Cu^+$ ions in solution; additionally, binding of $Cu^{2+}$ was demonstrated by the UV-visible spectra (Fig. 2A). Furthermore, the NMR data of H4pep indicate that $Zn^{2+}$ and $Cu^+$ binding occurs via the histidine side chains.

**Secondary structure and pH dependence of metal binding**
As anticipated by NMR, CD spectroscopy of H4pep in the far-UV range (190–250 nm) reveals only a deep band at 198 nm indicative of a random coil for the apo-peptide in buffer solution (Fig. 2B). Based on our peptide design strategy, we can reasonably anticipate that binding of $Cu^{2+}$ to H4pep will induce the formation of a β-sheet structural motif. Thus, we expect to observe spectral changes characteristic of β-sheet conformation when

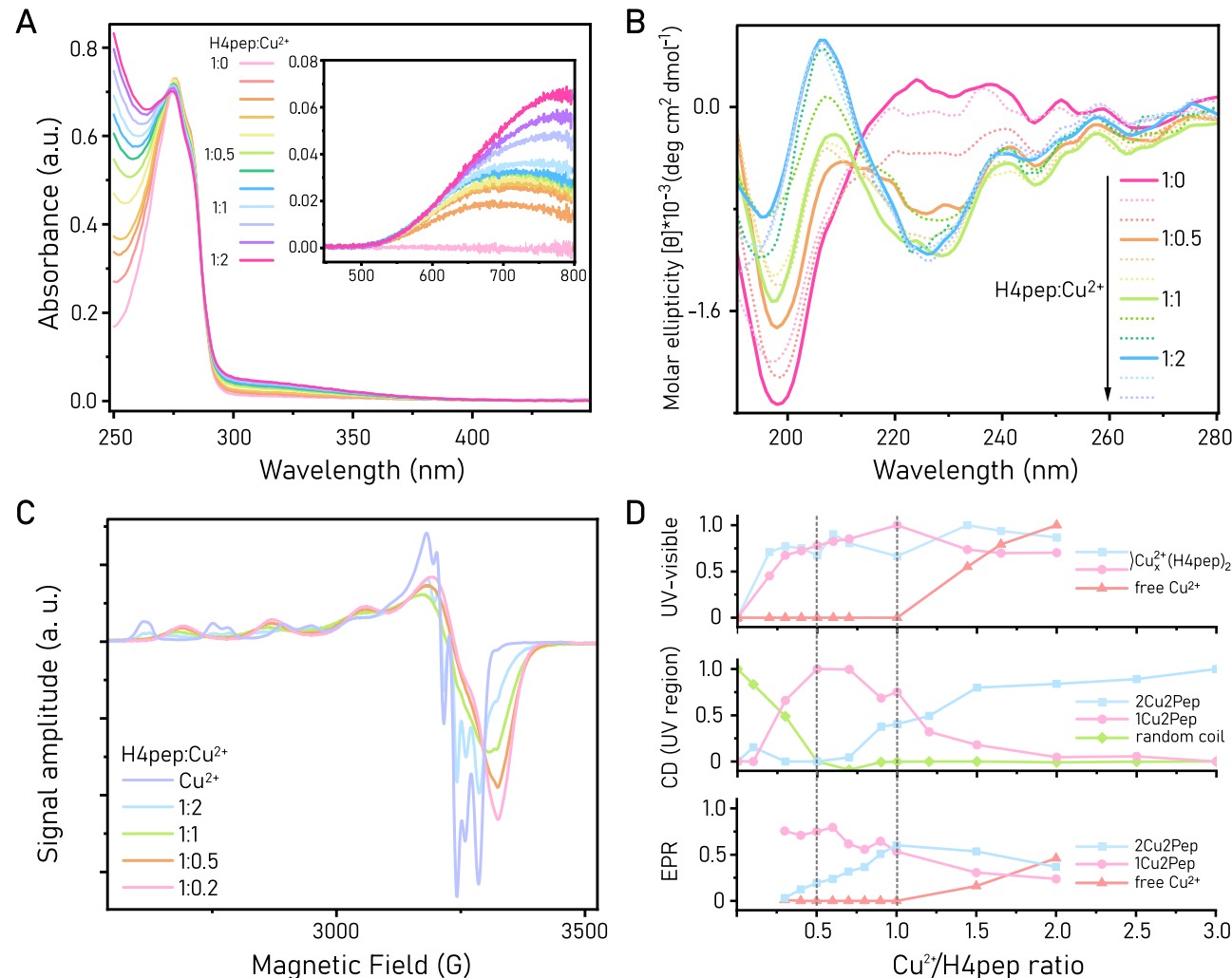

**Fig. 2 | Structural characterization of H4pep-Cu²⁺ species. A, B** UV-visible (**A**, [H4pep] = 0.4 mM, pH 5.6) and UV-CD spectra (**B**, [H4pep] = 0.1 mM, pH 5.6) of H4pep-Cu²⁺ complexes at increasing Cu²⁺ equivalents, ranging from 1:0 to 1:2 H4pep:Cu²⁺ ratios. Inset in (**A**) ([H4pep] = 2.35 mM), focuses on the Cu²⁺ d-d band in the range from 500 to 800 nm. Solid lines in (**B**) highlight significant samples in determining the stoichiometry of the complexes. **C** EPR spectra of Cu²⁺ and H4pep-

Cu²⁺ species at different H4pep:Cu²⁺ ratios (acetate buffer, pH 5.6). Spectra were recorded in presence of a fixed Cu²⁺ concentration (0.1 mM) and increasing concentrations of H4pep. **D** Speciation diagrams obtained by spectral fitting of UV-visible data and spectral deconvolution of CD and EPR data. **1Cu2Pep** and **2Cu2Pep** species are assigned according to the "Stoichiometry of H4pep-Cu²⁺ complexes" section.

H4pep is titrated with Cu²⁺. In line with our design and the previous NMR analysis, we expect metal binding to occur via histidine side chain coordination rather than backbone amide coordination, previously reported for 3-residue-Cu systems[35]. These modes of coordination can be differentiated by their pH dependence: histidine coordination is favored at lower pH due to the side chain imidazole group's pKa (~6), while backbone amide coordination typically predominates at higher pH (pKa ~7–8). Therefore, CD spectra recorded at different pH values allowed us to test Cu-binding ability under varying conditions. H4pep was dissolved in acetate buffer solutions at pH 4.4, 4.8, 5.2, and 5.6, respectively (Fig. S7). Higher pH values are excluded to avoid the formation of copper oxide species, limiting Cu²⁺ availability to the peptide ligand, and to mitigate terminal amine binding. No significant changes were observed in the CD spectra at pH 4.4 and 4.8 in presence of Cu²⁺, indicating unfavorable binding conditions due to the protonated state of the histidine ligands. In contrast, at pH 5.2 and 5.6, a clear change in conformation is visible. Therefore, pH 5.6 buffer conditions were selected to record the titration spectra in Fig. 2B. Upon Cu²⁺ addition, the CD spectrum displays a negative band at 227 nm and a positive one at 210 nm. This suggests that Cu²⁺ binding to H4pep induces a β-sheet conformation, consistent with our initial design. These features saturate upon the addition of 0.5 Cu²⁺ equivalents and remain essentially unchanged until a 1:1

H4pep:Cu²⁺ ratio. As additional copper equivalents are introduced, the spectrum undergoes further changes, with the positive feature increasing and shifting to 205 nm. This suggests the formation of another species, favored at high copper concentrations.

The same trend was observed in the visible range of the CD spectra, in which changes in the coordination environment around the metal ion(s) can be monitored (Fig. S8A). Titration of up to 1 equivalent of Cu²⁺ results in the gradual increase of a positive peak centered at 719 nm. At higher Cu²⁺ equivalents, the peak position shifts to the red, suggesting the formation of a second H4pep-Cu²⁺ species.

Titration of Cu⁺ to H4pep results in similar changes in the UV region of the CD spectrum (Fig. S8B). There are no observable features in the visible region of the same CD spectrum, consistent with the lack of d-d transitions in Cu⁺ complexes. This confirms that the observed changes in the UV region of the CD spectrum are associated with the coordination of Cu⁺ ions to H4pep to form H4pep-Cu⁺ complexes.

**Titration experiments via EPR**

EPR spectroscopy was employed to further analyze copper(II) binding and to elucidate the presence of different H4pep-Cu²⁺ species. Titration of H4pep into a Cu²⁺ solution revealed the presence of at least two distinct EPR

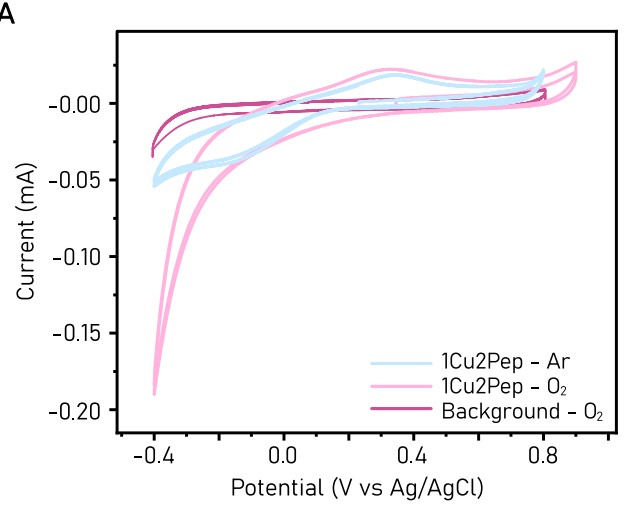

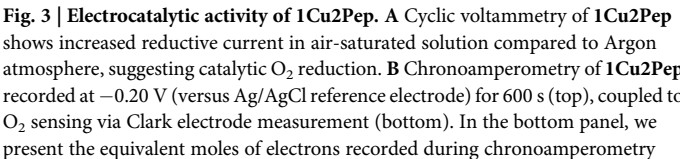

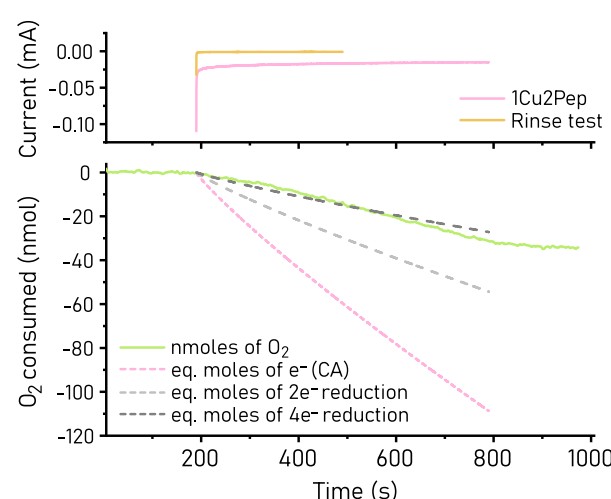

**Fig. 3 | Electrocatalytic activity of 1Cu2Pep. A** Cyclic voltammetry of **1Cu2Pep** shows increased reductive current in air-saturated solution compared to Argon atmosphere, suggesting catalytic $O_2$ reduction. **B** Chronoamperometry of **1Cu2Pep** recorded at −0.20 V (versus Ag/AgCl reference electrode) for 600 s (top), coupled to $O_2$ sensing via Clark electrode measurement (bottom). In the bottom panel, we present the equivalent moles of electrons recorded during chronoamperometry (CA) and alternative scenarios for two- and four-electron reduction of $O_2$. Rinse test chronoamperometry does not show significant reductive current, resulting in a flat trace in the Clark electrode response (not shown). All traces in (**A**) and (**B**) are recorded in 10 mM phosphate buffer with the addition of 40 mM $Na_2SO_4$ supporting electrolyte, in presence of 1 mM H4pep and 0.45 mM $Cu^{2+}$.

active species (Fig. 2C). Under a large excess of peptide (e.g., H4pep:$Cu^{2+}$ ratio 1:0.3), a single axial signal is observed, with values of $g_x$, $g_y$, and $g_z$, respectively, at 2.0419, 2.0737, and 2.2447 (Fig. S9). At higher $Cu^{2+}$ equivalents (between H4pep:$Cu^{2+}$ ratios 1:0.4 to 1:1) an additional axial signal characterized by lower g values could be detected. Presumably, the coordination of an additional $Cu^{2+}$ ion into the peptide can cause a small distortion to the local environment of the first $Cu^{2+}$, resulting in a shifted EPR spectrum, which could be isolated by spectral deconvolution (Fig. S10). In this configuration, the two $Cu^{2+}$ binding sites are essentially identical, as only one axial $Cu^{2+}$ signal is observed. The presence of two distinct EPR signals at different H4pep:$Cu^{2+}$ ratios is consistent with CD spectroscopy results, indicating that two H4pep-$Cu^{2+}$ species are formed. Furthermore, above H4pep:$Cu^{2+}$ ratio of 1:1, an EPR signal associated with unbound $Cu^{2+}$ becomes observable.

**Stoichiometry of H4pep-$Cu^{2+}$ complexes.** The CD and EPR spectra obtained from titrations between $Cu^{2+}$ and H4pep were deconvoluted by fitting each spectrum with its respective isolated EPR signals or CD spectra and presented in Fig. 2D. The observation of unbound $Cu^{2+}$ species in the EPR spectra above H4pep:$Cu^{2+}$ ratio of 1:1, suggests that each peptide can bind a maximum of 1 $Cu^{2+}$ equivalent. The formation of a β-sheet conformation upon $Cu^{2+}$ binding, as evident from the CD spectra, indicates that at least two peptide units are required to form a $Cu^{2+}_x(H4pep)_y$ complex (where y ≥ 2). It is unlikely that the minimum number of peptides to form a $Cu^{2+}_x(H4pep)_y$ is three, since we would expect most of the H4pep to exist in a β-sheet conformation at H4pep:$Cu^{2+}$ ratio of 1:0.33. On the contrary, our experimental observation led to a fitted molar fraction of peptide in random coil conformation of around 0.5 at 1:0.33 H4pep:$Cu^{2+}$ ratio (Fig. 2D). Therefore, our hypothesis is that a $Cu^{2+}_x(H_4pep)_2$ complex is formed upon addition of $Cu^{2+}$ ions to H4pep. In this scenario, there are two possible species corresponding to $Cu^{2+}_1(H_4pep)_2$ and $Cu^{2+}_2(H_4pep)_2$, designated as **1Cu2Pep** and **2Cu2Pep**, respectively. Deconvolution of UV-visible d-d band spectra could be obtained accordingly (Fig. S11). Fitting of the spectra up to 1 Cu equivalents was possible considering two Gaussian components centered at 636 and 742 nm, likely due to the formation of $Cu^{2+}_x(H_4pep)_2$ complexes. At higher $Cu^{2+}$ concentrations above 1 equivalent, a third component, centered at 786 nm and arising from free $Cu^{2+}$ in solution, needed to be included for accurate fitting.

Observing the resulting trends, it is clear how the isolation of a single $Cu^{2+}_x(H4pep)_2$ species is far from trivial. The **1Cu2Pep** species seems to be favored at low $Cu^{2+}$ concentrations, up to 0.5 $Cu^{2+}$ equivalents, while both **1Cu2Pep** and **2Cu2Pep** species are present in similar quantities up to 1 $Cu^{2+}$ equivalent. The **2Cu2Pep** species is favored at higher $Cu^{2+}$ concentrations, but an increasing amount of free $Cu^{2+}$ ions is detected in these conditions. Notably, even in large excess of $Cu^{2+}$, dynamic light scattering (DLS) experiments revealed the absence of large aggregates, confirming the molecular nature of the complexes (Fig. S12).

### Electrochemistry and testing of activity

To examine if **1Cu2Pep** and **2Cu2Pep** exhibit catalytic $O_2$ reduction activity, as in the original laccase, a preliminary electrochemical characterization of these two complexes by cyclic voltammetry was performed (Figs. 3 and S13A). Cyclic voltammograms of **2Cu2Pep** under $N_2$ atmosphere showed a reduction peak at −0.11 V against a Ag/AgCl reference electrode and a corresponding re-oxidation wave at 0.52 V. The large peak-to-peak separation suggests a slow heterogeneous electron transfer rate between the glassy carbon working electrode and the complex. Notably, during the anodic scan, a sharp reoxidation peak at 0.08 V is observed, characteristic of oxidation of metallic Cu deposited on the working electrode. This suggests that during the reduction of **2Cu2Pep**, the copper binding affinity of at least one of the binding sites becomes weaker, resulting in the loss of copper from the complex. Since metallic Cu is also known to participate in electrocatalytic $O_2$ reduction, we will not examine the $O_2$ reduction activity of **2Cu2Pep** as it is not trivial to verify its catalytic $O_2$ reduction activity in presence of metallic Cu.

On the other hand, cyclic voltammograms of **1Cu2Pep** show a reduction peak at −0.16 V and the corresponding reoxidation peak at 0.36 V under $N_2$ atmosphere. Importantly, features associated with the deposition of metallic Cu and its subsequent reoxidation are not observable. This is further verified by rinse test experiments performed after cyclic voltammetry of **1Cu2Pep** (Fig. S13B). Under $O_2$ atmosphere, a current enhancement is observable at around −0.20 V, suggesting catalytic $O_2$ reduction activity (Fig. 3A). To verify this hypothesis, bulk electrolysis experiments were performed in a custom-designed cell that incorporates a Clark electrode for sensing of $O_2$ concentration. Chronoamperometry experiments at −0.20 V result in a steady-state current of approximately −15 μA with a concomitant decrease in $O_2$ concentration (Fig. 3B),

**Fig. 4 | Plausible structures of 1Cu2Pep and 2Cu2Pep.** Optimized structures of **1Cu2Pep** (left) and **2Cu2Pep** (right). **2Cu2Pep** belongs to the $C_2$ point group, where the indicated $C_2$ rotation axis is perpendicular to the page. The two Cu binding sites are symmetrically equivalent.

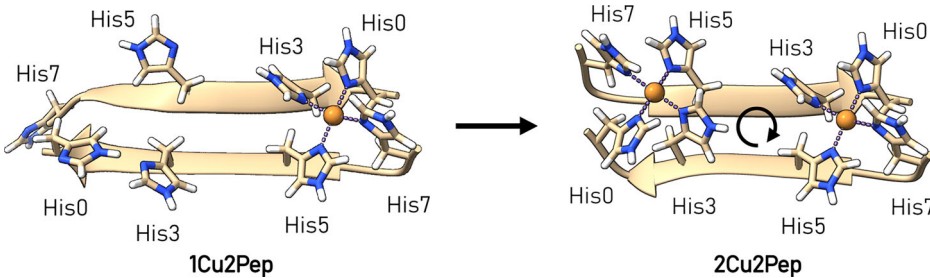

confirming that the catalytic current is due to $O_2$ reduction. At present, it is difficult to estimate a meaningful faradaic efficiency parameter, as the current data cannot distinguish between a two-electron reduction of $O_2$ to $H_2O_2$ or a four-electron reduction to water. As such, we present two scenarios to estimate faradaic efficiencies corresponding to two- or four-electron reduction of $O_2$ in Fig. 3B. In the four-electron reduction scenario, the faradaic efficiency approaches unity, while it is halved for the two-electron reduction scenario. We note that additional scenarios where mixtures of $H_2O_2$ and $H_2O$ products are also highly likely.

Cyclic voltammetry of the **1Cu2Pep** solution after bulk electrolysis does not show significant deviation from that recorded prior to bulk electrolysis, with no indication of $Cu^0$ plated on the electrode surface (Fig. S13C). Furthermore, additional rinse test experiments show a reductive current comparable to the background during chronoamperometry (Fig. S13D). To further rule out the participation of unbound copper species during catalytic $O_2$ reduction, negative control experiments were performed with dilute solutions of unbound $Cu^{2+}$ (0.03 mM, 7% of $Cu^{2+}$ content in **1Cu2Pep**). Chronoamperometry of **1Cu2Pep** shows a higher current compared to 7% $Cu^{2+}$ (Fig. S14A), indicating that free $Cu^{2+}$ ions only contribute minimally, if at all, to the catalytic activity observed for **1Cu2Pep**. Moreover, rinse test cyclic voltammogram of the unbound $Cu^{2+}$ solution shows significant reoxidation of metallic copper species at 0.09 V, not observed for the **1Cu2Pep** species (Fig. S14B). Therefore, the $O_2$ reduction activity in the conditions above is likely to originate from **1Cu2Pep**.

We have then tested the performances of **1Cu2Pep** at a more reducing potential (−0.30 V). In this case, an interesting behavior of the chronoamperometry signal was observed: while the recorded current decreases in the first 120 s, it then reaches a plateau and starts increasing after 150 s (Fig. S15A). No signals of unbound $Cu^{2+}$ could be detected in the cyclic voltammogram and rinse test performed after the measurement (Fig. S15B). This result seems to indicate that a more active species of the H4pep-Cu assembly can be formed in these conditions.

For preliminary assessment of $O_2$ reduction kinetics of **1Cu2Pep** and **2Cu2Pep**, we monitored the oxidation rate of p-phenylenediamine (PPD), commonly used in laccase activity assay, by spectrophotometry[36]. In a 1 mM PPD and 10 μM 1:1 $Cu^{2+}$:H4pep complex reaction mixture, the apparent reaction rate is 82.8 ± 36.6 nM min$^{-1}$. In the discussion below, we note that kinetic parameters derived from the use of typical laccase activity may not be representative of intrinsic reactivity.

## Discussion

Designing peptide mimics of enzymatic metal-binding sites is a challenging task. While most bioinformatics methods, such as Rosetta and Alphafold[37,38], emphasize a structure-based design of large proteins, our approach demonstrates that bioinformatic tools such as the here-proposed Metal-Site Analyzer can support the rational design of such mimics via analysis of sequence conservation. Short, tunable peptide ligands, among which α-helical structures are more widely adopted[7,9,10,29,39–42], represent promising candidates for applying this strategy. Notable examples are minimal Cu-peptide complexes, which are already broadly investigated since histidine-containing binding sites are found in several metalloproteins[10,43–46]. In these smaller systems, metal coordination often

involves backbone amide and terminal amine groups, as in the ATCUN motif[43,44,47]. In contrast, our H4pep-$Cu^{2+}$ complexes feature exclusive histidine coordination, as previously observed for copper-binding helical bundles[10,29,40,48,49], and for turn-forming and amyloidogenic short peptides[45,46,50,51].

At present, we are unable to directly characterize the structures of **1Cu2Pep** and **2 Cu2Pep** via X-ray crystallography or NMR. Instead, we performed geometry optimization using density functional level of theory (DTF) to derive plausible structures that are consistent with experimental NMR, EPR and CD observations (Fig. 4). At low copper concentrations, we hypothesize that $Cu^{2+}$ preferentially binds to a specific site, where the metal atom is bound to two peptide strands (**1Cu2Pep** species), as indicated by the formation of β-sheet motif observed in CD experiments. Assuming that the peptide strands associate in an anti-parallel β-sheet structure, two symmetrically equivalent binding sites can be identified. In this scenario, although only four out of eight available histidine residues coordinate $Cu^{2+}$ at a given time, all of them (His0, His3, His5, His7) are effectively involved in metal binding. This is reflected in the observed simultaneous broadening of all histidine proton resonances in the NMR experiments. As expected, a single axial $Cu^{2+}$ signal is observed in EPR experiments, corresponding to a mononuclear Cu species. At higher copper concentrations, coordination of a second $Cu^{2+}$ can occur, filling the second binding site. However, this leads to a small distortion of both binding sites. The resulting **2Cu2Pep** species features two symmetrically identical binding sites, related by a $C_2$ rotation axis as indicated in Fig. 4. The two identical Cu sites of **2Cu2Pep** differ slightly from the Cu site of the **1Cu2Pep** species. Therefore, the EPR signature of **2Cu2Pep** displays a single axial copper signal, which is only slightly shifted compared to that of the **1Cu2Pep** species, as the two identical $Cu^{2+}$ ions are not magnetically coupled (Figs. 2C and S10B). The NMR spectra in a 1:1 H4pep:$Cu^+$ ratio also show no significant differences compared to the 2:1 H4pep:$Cu^+$ ratio, supporting our interpretation (Figs. S5 and S6).

The computational modeling and spectroscopic characterization of 1Cu2Pep and 2Cu2Pep reveal a controlled assembly process yielding stable metallo-β-sheet structures with well-defined metal coordination environments. Only recently, metallo-β-sheets have been considered as scaffolds for catalytic site mimics, with just a handful of examples[50,52–55]. A significant challenge in this field is that β-sheet motifs typically self-assemble into supramolecular structures and form insoluble aggregates[6,54,56]. Our work demonstrates how metal coordination can be leveraged to control β-sheet assembly, preventing undesired aggregation. This controlled assembly represents a significant advance in rational metallo-β-sheet design, providing molecular-level insight into the factors governing metal-mediated peptide association. This understanding of sequence-structure-metal binding relationships is crucial for utilizing these structural motifs across various research fields, including the study of amyloid-β aggregates implicated in neurodegenerative disorders such as Alzheimer's disease[57,58]. Our success in reproducing a molecular metallo-β-sheet motif via bioinformatic and structural design highlights the utility of MeSA and provides a step forward in understanding the factors governing the stability and assembly of this important structural motif.

Our preliminary chronoamperometry experiments, in combination with in situ $O_2$ concentration measurements, verified that the enhanced current observed in cyclic voltammograms of **1Cu2Pep** under $O_2$ atmosphere is due to catalytic $O_2$ reduction reactions. As mentioned above, it is not possible to assign the catalytic $O_2$ reduction to a two- or a four-electron reduction process with the present data. While it is tantalizing to expect four-electron reduction to water, approaching 100% faradaic efficiency, this raises questions regarding the mechanism by which a mononuclear Cu complex could facilitate such reaction. This scenario implies the presence of a high-valent square pyramidal or tetragonal copper(III)-oxo species, seemingly in violation of the 'oxo-wall' principle[59]. However, it has been proposed that this can be circumvented by structural rearrangement, forming a tetrahedrally distorted N3-Cu(III)-oxo complex, thus allowing for subsequent reduction to achieve four-electron reduction to water[60]. Alternatively, four-electron reduction of $O_2$ can proceed via a bimolecular pathway, with $O_2$ bridging two **1Cu2Pep** complexes. While these scenarios do not constitute definitive proof of four-electron reduction of $O_2$ to water by **1Cu2Pep**, they underly the plausibility of such pathways.

During our investigation using laccase substrate analogues, we note that the observed reaction rate for PPD oxidation is very low and remains constant over 15 min. Such a low reaction rate is likely due to coulombic repulsion between the positive charge of the metallo-peptide complex and the positively charged PPD at pH 5.6[61,62]. On the other hand, the anionic 2,2′-azinobis(3-ethylbenzthiazoline-6-sulfonate) (ABTS), another common substrate for laccase activity, interacted strongly with our metallo-peptide complexes, precluding any meaningful activity measurements. Our observations suggest that the use of such assays for determining the catalytic activity of synthetic mimics of laccase is susceptible to bias due to the selection of laccase substrate analogues. As such, we cautiously refrain from comparison of the catalytic performance of **1Cu2Pep** and **2Cu2Pep** based on these assays. Nonetheless, based on the present activity studies, we can conclude that **1Cu2Pep** and **2Cu2Pep** show positive catalytic activity for $O_2$ reduction.

Despite some noteworthy examples of molecular mononuclear copper complexes[63], there are only a few examples of mononuclear copper-peptide complexes reported to be active towards dioxygen activation in the literature[27]. Ascorbate oxidase activity has been observed for some ATCUN sites, particularly when histidine is not located in the N-terminal position[64,65], often accompanied by catalytic self-oxidation[66]. However, no quantitative catalytic parameters have been directly reported. Interestingly, modest oxidase activity towards o-phenylenediamine has been observed for a terminal His residue covalently linked to the N of a Lys side chain via an isopeptide bond, reminiscent of the coordinating motif found in lytic polysaccharide monooxygenases (LPMOs)[67]. Finally, Makhlynets et al. reported initial oxidase activity rates toward 2,6-dimethoxyphenol for a library of short amyloidogenic copper-binding-peptides in the range 0.5–5 $M min^{-1}$. Their study highlighted the importance of residue identity in specific positions by keeping constant the number of His residues and their relative position in the sequence[50].

Although the observed catalytic activity of **1Cu2Pep** and **2Cu2Pep** appears very modest compared to their parent laccases, these complexes serve as proof-of-concept that structurally simple systems formed by minimal peptide fragments can recapitulate some functional features of natural enzymes. Further efforts could focus on designing peptide sequences that self-assemble into a single, well-defined metallo-peptide complex. In this context, more rigid motifs, such as β-hairpins and WW domains, have recently been proposed as more stable architectures for achieving this goal[68–70]. However, recent work from Dang et al. has demonstrated that β-hairpin structures are particularly affected by metal binding, which can cause a rearrangement of the β-sheet domains[46]. Nonetheless, mutating just two amino acids in the sequence resulted in a completely different metal-peptide assembly mechanism, underlining the critical importance of sequence analysis in designing selective and stable binding sites. The proposed MeSA tool plays a central role in the design of these β-sheet-forming peptide fragments, which coordinate copper ions on an accessible face. We further anticipate that the use of MeSA can be generalized to enable the identification and rational design of short metallo-peptides with predefined structural motifs, encapsulating salient features related to the structure and activity of the parent metalloprotein. Such well-defined and minimal architectures make our MeSA tool a versatile platform for probing structure–function relationships in a controlled and modular way.

## Methods

### Chemicals and buffers

All chemicals and reagents were of analytical grade and used without further purification. Fmoc-amino acids and Oxyma pure were purchased from Novabiochem (Sigma-Aldrich Sweden). Dimethylformamide (DMF), 20% piperidine, N,N′-diisopropylcarbodiimide (DIC), trifluoroacetic acid (TFA), triisopropylsilane (TIS), diethyl ether, acetonitrile (ACN), copper(II) sulfate pentahydrate, sodium dihydrogen phosphate, sodium hydroxide (NaOH), sodium acetate, acetic acid, sodium sulfate ($NaSO_4$) and 2,2′-azino-bis(3-ethylbenzthiazoline-6-sulphonic acid) (ABTS) were purchased from Sigma-Aldrich Sweden. Buffers were prepared starting from sodium dihydrogen phosphate and adjusting the pH with NaOH, or mixing sodium acetate and acetic acid to the desired pH. $Cu^{2+}$ stock solutions were obtained by dissolving copper sulfate salt in distilled water or buffer.

### Bioinformatic tool

Metal-Site Analyzer, available at https://metalsite-analyzer.cerm.unifi.it/, was designed and implemented with the following characteristics. The input consists of a pdb code, or a protein structure in pdb format (e.g., in case the structure is not yet deposited in the PDB). It is also possible to specify the distance threshold that will be used to identify the coordinating atoms to the metal ion(s) and the residues directly involved in the metal binding. If the structure contains a multinuclear site, the user can choose whether to analyze a single metal ion in the site or investigate the entire multinuclear site. In the latter case, it is required to flag the "Aggregate multinuclear site info" option. Then, by clicking on "Search for metals", the tool extracts the list of metal sites in the structure. If the user prefers to limit the search to a specific metal ion, this can be specified in the "Enter a Cofactor" field on the home page of the tool. The second web page displays the list of sites found in the input structure. The user must select only one site of interest. The "GO!" button will then initiate the site analysis (Fig. 5) through the steps listed below:

1. The tool extracts the "minimal functional site" (MFS) from the structure[14]; that is, (i) the metal(s), (ii) the metal-binding residues, and (iii) the residues that fall within 5 Å from at least one atom of the metal-binding residues. Usually, the MFS is composed of more than one fragment (*metal-binding fragments* hereafter), because the metal site is not continuous in sequence.
2. The residues identified at points (ii) and (iii) are mapped onto the sequence of the structure.
3. The tool extracts the segment of sequence(s) that contains the entire MFS (from the most N-terminal residue to the most C-terminal residue) (*metal-binding segment* hereafter).
4. The metal-binding segment is searched in UniRef50 by using PSI-BLAST (3 iterations, using default parameters of PSI-BLAST)[19,20]. UniRef50 was chosen because it contains a sub-selection of the whole UniProt knowledgebase at the level of 50% sequence identity, thereby reducing its degree of redundancy (approximately only 21% of the proteins in UniProt are retained in UniRef50)[71].
5. The output sequences are filtered to discard those not containing the metal-binding residues of the input site (as they will most likely not bind metal ion(s)). The number of retained sequences is displayed in the output page.
6. A multiple sequence alignment is generated from the pairwise sequence alignments of the PSI-BLAST output.
7. The metal-binding fragments of the input site are mapped and extracted from the multiple sequence alignment.

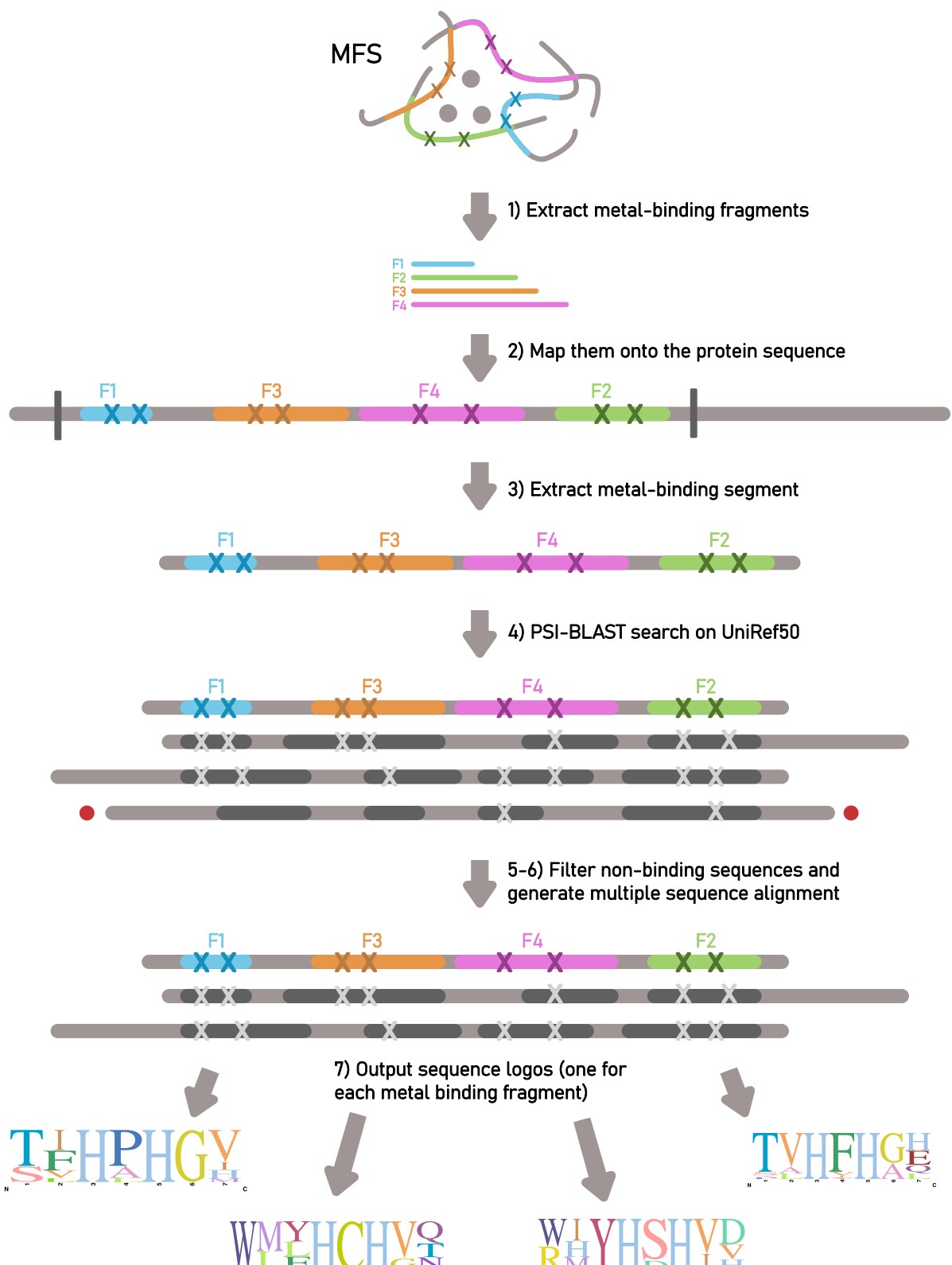

**Fig. 5 | Schematic representation of the MeSA workflow.** At the top, a hypothetical MFS is displayed, with binding residues marked with an "X". Metal-binding fragments are indicated as F1, F2, F3, and F4. The scheme illustrates the workflow of MeSA as described in the "Bioinformatic tool" section. Output sequence logos are also hypothetical representations.

At the end of the site analysis, the web server will output the metal-binding fragments alignments and the respective Skylign logos[26]. The latter may be used to identify highly conserved residues, which may have a functional role in the metal site and thus be relevant to design purposes.

**Peptide design**

The design of the synthetic peptide was based on the small laccase from Streptomyces viridosporus with PDB code 3tbc. The PDB structure was input to MeSA with no additional constraints. The option to aggregate

multinuclear site information was chosen to be able to include the analysis of the trinuclear copper site of interest. The tool individuated three mono-nuclear type 1 copper sites, respectively belonging to the three monomers that compose the protein (chains A, B, and C), and three trinuclear sites, located at the interface between monomers. All sites are virtually identical, as the monomer unit repeats with a three-fold symmetry in the functional protein structure. By selecting one of the trinuclear MFS, four fragments were extracted, two belonging to one chain (TFHLHGH and WMYHCHVQSHS) and two to another (SLHVHGLDY and WHYHDHVVGTEHG). By looking at the three-dimensional structure of the MFS, we noticed that fragments on the same chain are part of adjacent beta-strands located at the interface of one monomer, while the other fragments (on the second chain) bind the Cu ions from the opposite monomer interface, in a similar beta-sheet conformation. The binding motif His-Xxx-His is highly conserved in all four fragments, as suggested by the MeSA output. For our purpose of designing a minimal-length peptide to mimic the laccase binding site, we have focused on the sequence con-servation analysis of the shortest fragments, respectively, TFHLHGH belonging to one chain and SLHVHGLDY belonging to the other. The highly conserved binding histidine residues were selected for positions 3 and 5, as well as the glycine in position 6 (Fig. 1B). A third histidine was assigned to position 7, as it is fairly conserved and capable of also participating in metal binding. Flexible threonine was chosen versus the more rigid proline and assigned to position 1, as the third and second most conserved residue in the two fragments, respectively. The other positions seemed to be less conserved among the fragments and were assigned based on the analysis of the three-dimensional structure of the model site. Tyrosine was strategically included in the sequence as it can be used as experimental UV-visible and redox probe. The final sequence HTVHYHGH is an 8-residue-long peptide representing the shortest frag-ment mimicking the model site and containing the conserved binding motif. It is important to note that in the MFS of laccase, each Cu atom is coordi-nated by histidine residues from different fragments; to explore the feasi-bility of copper binding with a H4pep:Cu ratio of 1:1, an additional His residue at position 0 was introduced to satisfy the requirement of a four-coordination environment.

## Peptide synthesis

Peptide synthesis was performed via Fmoc-chemistry using an automated microwave peptide synthesizer (Biotage Initiator + , Alstra) on a 0.1 mmol scale using Rink-type resin (TentaGel® R RAM) with a polyethylene glycol/polystyrene backbone. The amino acids (Fmoc-His(Trt)-OH, Fmoc-Gly-OH, Fmoc-Tyr(tBu)-OH, Fmoc-Val-OH, Fmoc-Thr(tBu)-OH) were weighed in concentrations of 0.5 M and dissolved in DMF. 20% piperidine in DMF was used as the deprotection solution. All coupling steps were performed at 75 °C for 5 min, except for the His residues, which were coupled at room temperature for 60 min to avoid racemization. Oxyma (0.5 M in DMF) and DIC (0.5 M in DMF) were used as the activator and activator base, respectively. Final deprotection was performed after the last coupling cycle to obtain a free amine terminus. Cleavage of the peptides from the resin and side chain deprotection was achieved using a 95% trifluoroacetic acid TFA, 2.5% TIS, and 2.5% distilled water cleavage cocktail for 2 h at room temperature with stirring. After cleavage, the crude peptides were filtered, and excess TFA was evaporated under a gentle stream of $N_2$ gas. The peptides were precipitated, washed with cold diethyl ether, dissolved in water, and lyophilized. The peptides were dissolved in water for purification over a C18 semi-preparative column using an ÄKTA pure chromatography system (Fig. S16). Solvent A was 0.1% TFA in $H_2O$, and solvent B was 0.085% TFA in acetonitrile. The peptide was injected in the equilibrated column with 5% B for 5 min, fol-lowed by a linear gradient to 20% B over 30 min at a 4 mL min$^{-1}$ flow rate. The peptide was eluted at 10% solvent B. The chromatogram was monitored at 280 nm. The purified peptide was lyophilized and stored at −20 °C. Peptide mass and purity were analyzed by LC-MS (Agilent Technology) (Fig. S17).

## Spectroscopic methods: UV-vis, CD, EPR, NMR

UV-vis spectra were recorded using a Cary 50 Bio (Varian) spectrometer, in a 10 mm path-length disposable cuvette containing 10 mM acetate buffer at pH = 5.6. The peptide concentration for the near-UV range was 0.4 mM, with addition of 20 μL aliquots of $Cu^{2+}$ from a 1 mM copper sulfate stock solution. For the visible range spectra, a peptide concentration of 2.35 mM was used, and a $Cu^{2+}$ stock concentration of 22.8 mM.

CD spectra were recorded with a Chirascan V100 spectrometer (Applied Photophysics) equipped with a temperature-controlled cell Peltier holder. Smoothing via adjacent averaging was applied when indicated. The spectra were converted from millidegrees to units of mean residue molar ellipticity [Θ] in deg cm$^2$ dmol$^{-1}$ by setting $[\Theta] = 100*\theta_{obs}/Cln$, where $\theta_{obs}$ is the recorded ellipticity in millidegrees, $C$ is the concentration of protein in mM, $l$ is the cuvette pathlength in cm, and $n$ is the number of amino acids in the sequence ($n = 8$). For UV-range measurements, a 1 mm path-length quartz cuvette was used. The peptide was dissolved in 10 mM acetate buffer at pH 5.6 to obtain a 0.1 mM solution. $Cu^{2+}$ aliquots were added from a 1 mM $Cu^{2+}$ stock solution. For visible-range measurements, a 10 mm path-length quartz cuvette was used, and the sample was prepared with a peptide concentration of 1 mM and a 10 mM $Cu^{2+}$ stock solution. In addition, spectra were recorded in 10 mM NaPi buffer solution at pH 5.8 and displayed no significant changes.

All EPR spectra were recorded on a Bruker EMX-micro spectrometer equipped with an EMX-Premium bridge and an ER4119HS resonator in connection with an Oxford Instruments continuous flow cryostat. Low temperatures were reached using liquid helium flow through an ITC 503 temperature controller (Oxford Instruments). The EPR samples were pre-pared in 10 mM acetate buffer at pH 5.6 with a fixed $Cu^{2+}$ concentration of 0.1 mM and different peptide concentrations, to reach the selected sto-chiometric ratio. The tubes were flash-frozen and stored at 77 K prior to EPR analysis. The EasySpin software version 6.0.5 was used for spectral simulation and fitting[72,73]. The spectra were recorded at 10 K, microwave power of 50 μW, microwave frequency of 9.31 GHz, modulation amplitude of 10 G, and modulation frequency of 100 MHz.

NMR spectra were acquired at 298 K on a Bruker Avance 600 Spec-trometer equipped with a triple-resonance cryoprobe. Samples for NMR analysis were prepared by dissolving H4pep in either sodium phosphate buffer pH 5.8 (10% $D_2O$) or $CD_3OH$ to a final concentration of 0.5 mM. For metal-binding studies, $ZnCl_2$ or $[Cu(CH_3CN)_4]PF_6$ was added in stoi-chiometric amounts. For $Cu^+$ experiments, samples were prepared with deoxygenated buffer and kept under inert Ar atmosphere. Sodium dithio-nite (3 mM final concentration) was added to maintain reducing conditions. We have not observed any significant difference in NMR spectrum before and after addition of excess of dithionite, therefore excluding dithionite binding or copper oxidation during spectra acquisition. Water suppression was accomplished using an excitation sculpting sequence[74]. Chemical shifts are referenced to TSP.

## Competitive titration studies

$Cu^+$ binding affinity was determined through competitive titration experiments with BCA (bicinchoninic acid) as a competing ligand ($\log\beta_2 = 17.2$)[75]. Experiments were performed by adding small aliquots of BCA (7.2 mM stock solution) to a solution containing H4pep-$Cu^+$ complex (prepared with 0.95 equivalents of $[Cu(CH_3CN)_4]PF_6$ dissolved in acet-onitrile up to 10 mM concentration) in 50 mM sodium phosphate buffer at pH 5.8 with 1 mM sodium ascorbate under inert atmosphere. Formation of $[Cu(BCA)_2]^{3-}$ was monitored spectrophotometrically at 562 nm ($\varepsilon_{562} = 7900$ M$^{-1}$ cm$^{-1}$). The dissociation constant was determined by fitting the titration data to equation[75]:

$$[L]_{tot} = 2[ML_2] + \sqrt{\frac{[ML_2]}{K_D\beta_2}\left(\frac{[P]_{tot}}{[M]_{tot} - [ML_2]} - 1\right)}$$

Where $[L]_{tot}$, $[P]_{tot}$, $[M]_{tot}$ are the total BCA, peptide, and $Cu^+$ concentra-tions, respectively, and $[ML_2]$ is the $[Cu(BCA)_2]^{3-}$ complex concentration.

## Electrochemical measurements and activity assay

All experiments were performed with a Metrohm Autolab potentiostat or a BioLogic SP-300 potentiostat at room temperature, using a low-volume electrochemical cell. For CV measurements, a 3 mm diameter GCE working electrode was used. For bulk electrolysis measurements, the diameter of the exposed working electrode area was 9 mm. A graphite rod was used as counter electrode. An Ag/AgCl (in 3 M KCl) electrode was used as a reference and regularly checked using $K_4[Fe(CN)_6]$ as standard ($E°_{Ag/AgCl}$ = +0.210 V vs SHE). All potentials are reported versus Ag/AgCl. For all measurements, the glassy carbon working electrodes were polished with 0.05 μm alumina particles and sonicated for 3 min in distilled water immediately before use. Measurements were performed in 10 mM sodium phosphate buffer at pH 5.8, in addition of 40 mM $Na_2SO_4$ as supporting electrolyte. **1Cu2Pep** cyclic voltammetry was performed in presence of 1 mM H4pep and 0.45 mM of $Cu^{2+}$, while **2Cu2Pep** measurements were performed in presence of 1 mM H4pep and 0.9 mM of $Cu^{2+}$. The total solution volume was 3 mL. Activity measurements were recorded by integrating a custom-made Clark-type electrode connected to a Unisense control box. Rinse test measurements were performed immediately after the voltammetry/amperometry by removing the active solution, rinsing the electrode with distilled water for 10 seconds, and reassembling the cell with fresh buffer and supporting electrolyte.

Laccase activity was determined by spectrophotometric assay by measuring the oxidation of 1 mM PPD ($\varepsilon_{530} = 7650\ M^{-1}\ cm^{-1}$) in sodium acetate buffer (0.1 M at pH 5.6), and of 0.5 mM ABTS ($\varepsilon_{4290} = 3600\ M^{-1}\ cm^{-1}$) in phosphate buffer (20 mM at pH 6). Sample aliquots were prepared by mixing a 1:1 ratio of H4pep and $Cu^{2+}$ stock solutions in the same buffer to achieve a final concentration of 10 μM. The formation of the product was measured over 15 min by recording the change in absorbance at 530 nm in the case of PPD, and for 5 min at 420 nm for ABTS. The activity assay was performed in triplicate.

## Structural prediction by computational chemistry

Plausible structures of **1Cu2Pep** and **2Cu2Pep** are obtained by geometry optimization of initial guess structures that are constructed based on information derived from our spectroscopic investigations. For computational efficiency, geometry optimization of initial guess structures is first performed at semi-empirical extended tight binding (GFN2-xTB) level of theory before final geometry optimization at DFT level of theory using the r2SCAN-3c composite method[76,77]. All calculations are performed with xTB 6.7 and ORCA 6.0 computational packages[78,79].

## Data availability

All data are available in the main text or the supplementary materials. Coordinates from DFT calculations are available as Supplementary Data 1 and Supplementary Data 2. NMR data are available as Supplementary Data 3. All data are available from the corresponding author upon request.

## Code availability

The Metal-Site Analyzer is available as a publicly accessible web interface at https://metalsite-analyzer.cerm.unifi.it/. The code for Metal-Site Analyzer is available from Claudia Andreini upon request.

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

## Acknowledgements

Swedish Energy Agency grant 50529-1 (C.S. and M.H.C.). Italian MUR, Project SEA-WAVE 2020BKK3W9 [CUP_E69J22001140005] (M.C., O.M. and A.L.). The computations were enabled by resources provided by the National Academic Infrastructure for Supercomputing in Sweden (NAISS), partially funded by the Swedish Research Council through grant agreement no. 2022-06725, project NAISS2024/22-1095.

## Author contributions

Conceptualization: C.S., C.A. and M.H.C. Methodology: C.S., C.A., M.C., A.L. and M.H.C. Investigation: C.S., M.C., O.M., P.H., C.A. and M.H.C. Supervision: A.L., C.A. and M.H.C. Writing—original draft: C.S. and M.H.C. Writing—review & editing: C.S., M.C., A.R., L.D.A., A.L., C.A. and M.H.C. with input from all authors.

## Funding

## Competing interests

The authors declare no competing interests.
