## [Transparent Peer Review file · Communications Chemistry]

A bioinformatics approach to design minimal biomimetic metal-binding peptides

Corresponding Author: Dr Mun Hon Cheah

Version 0:

Reviewer comments:

Reviewer #1

(Remarks to the Author)

Spallacci et al. report a laccase developed using a bioinformatic approach. Overall, this is an interesting piece of work that is of interest so ultimately I would support its publication. However, there are some issues that I'd like to get addressed before publication.

1. The website developed is a bit of a black box. The paper does not specify what exactly how exactly the databases are searched, how many sequences are analyzed, how redundancies are removed, etc. This is important for evaluating this work
2. Kinetic characterization. I feel U/L is not ideal (or meaningful) units. I appreciate the difficulties in establishing purely mol based parameters such as k_{cat} , K_M , etc. but U/L is not useful when comparing this work to that of others.
3. The speciation is a problem. The authors are fairly upfront about the mixed species formation but the overall conclusion need to explicitly address this issue. If the sequences derived from natural sources are so specific, how come such complex speciation is present?
The authors report a difference in conformation between 2:1 and 1:1 peptide:copper ratios. Does this affect the reactivity of the species? Can you relate the reactivity to the changes in character? The thesis depends on this being a beta-sheet protein with metal centres on the face, and given the structure of the baseline laccase, I feel like it begs the question of what precisely the requirements for kinetic activity are.
4. A question I have always had relates to the specificity of these types of catalysts and whether it is specific to these sequences or simply due to the promiscuity of coordinated copper- the authors mention similarity to Pecoraro's coiled coil nitrite reductases. Do these compounds specifically carry out laccase activity (not requiring hydrogen peroxide) or are they promiscuous? It would be interesting to see the reaction scope with different classic laccase characterizations. There are quite a few classic laccase substrates, of which ABTS is only one. <https://www.mdpi.com/2309-608X/7/2/143> is an excellent resource, and perhaps the authors might consider another common, inexpensive substrate like p-phenylenediamine to determine whether this is a laccase mimic, or simply another ABTS catalyst. Alternatively, they might study the 2,4-dichlorophenol/4-aminoantipyrine reaction that has been reported by several groups reporting laccase mimics. Gazit and co-workers recently reported a $Cu(Phe)_2$ laccase species with excellent catalytic parameters, and so the authors might wish to compare their protein-based result.
5. Do the authors observe turnover, in other words is there true catalysis?
6. the authors report the final design as an amalgam of the two fragments they report from Chain A and Chain B of their analysis, however, several elements are confusing to me-The first histidine seems to have been added to mimic the laccase metal site, the second threonine residue comes from selecting the next most conserved residue which is not proline, but I do question the design choices which led to position 4. As far as I can see from the figures, tyrosine is not conserved at all in this position. Did the authors try the different substitutions such as leucine, tryptophan, methionine, and others and not report them? How much "directed evolution" knowledge can we derive from these suggested sequences? It would be interesting to see how general this method is, as it is difficult to rationalize this as a metal binding site predictor should the system require a modification which is not present in the output. A true test of this design would be to use only residues predicted in the alignment.

Minor issues:

Statistical parameters are not always given

The “European” way of using comma as opposed to period in e.g. 10.0 is found in many places

Reviewer #2

(Remarks to the Author)

Major comments:

- 1) Designing a new tool MeSA to design peptides based on minimal functional site of a protein is exciting, however the reviewer could not use the stated link <https://metalsite-analyzer.cerm.unifi.it/>
- 2) Line 162: “pH 5.6 to prevent copper oxide formation and N-terminal amine deprotonation (vide infra), still favoring imidazole deprotonation and binding (pKa ~6).” At pH 5.6 most His would be protonated and not be a good ligand for metal. Also pKa of His would depend a lot on peptide sequence. It is easy to test how much metal ion is binding by running simple ITC experiment. Also CD experiments were performed at pH 4.4, 4.8, 5.2, and 5.6, not much binding of copper would be expected at low pH. Binding of terminal amine to Cu(II) is a valid concern, so protecting NH₂ with acetic anhydride would help to eliminate the binding.
- 3) For NMR experiment with Cu(I) how the samples were kept under inert atmosphere? Were they dissolved in a degassed buffer? How much dithionite was used? Was the sample in the NMR tube with a screw cap to prevent oxygen from getting in. It is important to see the details as oxidation of copper is possible and then copper could bind to dithionite and Cu(II) would result in broadening of NMR spectra.
- 4) Copper (II) in acetate buffer of 10 mM concentration would be complexed with acetate?

Minor:

- 1) Line 93: Our data demonstrate 92 that, despite its beta-sheet conformation, Cu₂+(H₄pep)₂ does not undergo aggregation, such that we 93 did not detect the formation of fibrils. No aggregation is based on what experiments?
- 2) Figure 2A: different color spectra correspond to how many equivalent of copper. Band at 786 nm was assigned to free Cu(II), why there is an increase in absorbance when amount of Cu(II) is low (presumably most ions will be bound to peptide)...
- 3) Fig. S1. How much copper was used, buffer pH. These values should be present in figure caption.
- 4) In Figure 1B, what is 1 and 2? It is in the main text, but not figure caption.
- 5) No glycerol was used to prepare EPR samples?
- 6) How oxidation state of copper is controlled in CD experiments?
- 7) Fig. S10: Stoichiometry – I is missing

Reviewer #3

(Remarks to the Author)

This manuscript presents a nice study on the design of a minimal biomimetic peptide that mimics the trinuclear copper site of laccase using a bioinformatics-driven approach. The authors describe a novel computational tool, MetalSite-Analyzer (MeSA), to extract metal-binding motifs from metalloenzymes, leading to the rational design of an eight-residue peptide (H₄pep) that binds Cu²⁺ and demonstrates catalytic activity. The work is well-motivated and methodologically sound, contributing to the field of bioinspired catalyst design.

However, the design is based on a 4 peptides:3 Cu active site, yet the stoichiometry observed and the model proposed in Fig S16 are quite different. In that sense, the design is not successful. What is the proposed mechanism for the observed activity?

- Did the authors explore alternative sequences with different residues in variable positions? This would help establish whether H₄pep is an optimized solution or simply one possibility.
- The EPR analysis, while demonstrating two distinct Cu²⁺-peptide species (1Cu₂Pep and 2Cu₂Pep), would benefit from a deeper discussion on why only a single axial signal is observed despite presumed symmetry distortions.
- The CD data indicate β-sheet formation upon Cu²⁺ binding, but no direct structural characterization (e.g., X-ray crystallography or NMR-derived structure) is provided. Can molecular dynamics simulations or secondary-structure predictions further support this conclusion?
- DSL measurements at high concentration of peptide show no aggregates, but are also nonreproducible. Have the authors tried sedimentation ultracentrifugation or SEC? How about AFT or TEM, which might capture structures relevant to the observed electrochemical activity?
- The electrocatalytic activity of 1Cu₂Pep is intriguing but remains modest in comparison to natural laccases.
- The faradaic efficiency (62%) suggests that side reactions (possibly O₂ reduction to H₂O₂) may be occurring. Have the authors tested for peroxide formation?
- At -0.30 V, the increasing current suggests an alternate active species forming over time. Can spectroscopic or electrochemical evidence confirm this?
- The cyclic voltammetry rinse tests suggest no metal deposition for 1Cu₂Pep, but what about long-term stability? Have the authors examined whether repeated catalysis deactivates the peptide?
- The study would benefit from a comparison with other reported biomimetic metal-peptide systems. For instance:
 - How does the catalytic efficiency of 1Cu₂Pep compare with previously reported Cu²⁺-peptide complexes?
 - Could a comparison to engineered metalloproteins or synthetic small-molecule catalysts provide better context for H₄pep's catalytic performance?

- The figures in the main text and supplement are generally clear, but some could be improved:
- Figure 2D (speciation diagram) would benefit from an explanation of how relative species distributions were derived.
- Supplementary Figure S11 (UV-Vis deconvolution) should include additional discussion on fitting assumptions.
- EPR simulations (Fig. S9–S10) should be described in more detail for readers unfamiliar with spectral deconvolution.
- The introduction is well-written but somewhat lengthy. Consider streamlining background information and moving some discussion on metalloenzyme active sites to the discussion section. This is especially in view of the fact that the design of a mini-laccase is not successful.
- The discussion sometimes reiterates results rather than providing deeper insights. A more concise, focused discussion on broader implications (e.g., potential for enzyme-inspired catalysis in industry) would improve readability.
- Minor grammatical errors are present. For example:
- “A promising strategy for designing biomimetic catalysts holds on mimicking...” → “A promising strategy for designing biomimetic catalysts relies on mimicking...”
- “Nature-inspired or biomimetic catalyst aims to reach”... should be “catalysts aim...”

Version 1:

Reviewer comments:

Reviewer #1

(Remarks to the Author)

The manuscript is improved enough to warrant publication in the present form.

Reviewer #2

(Remarks to the Author)

Most of the questions were addressed, however there remain several points that need further explanation.

1. The authors look at binding of Cu(I) to BCA, however Cu(II) is binding to peptides and not Cu(I), so I am confused by why binding affinity for Cu(I) was measured.

2. For the NMR experiment, Zn(II) is used as a model for paramagnetic Cu(II) and conclusions are made for which ligands bind to the metal ion. But Zn(II) is not a great substitute for Cu(II) as it prefers different ligands and coordination geometry (Cu(II) is octahedral and Zn(II) is tetrahedral, Zn(II) has more preference for S compared to Cu(II), which prefers N ligands). Did anyone report use of Zn(II) as a substitute for Cu(II)?

Version 2:

Reviewer comments:

Reviewer #2

(Remarks to the Author)

I reviewed the responses to my earlier comments and this that the paper is ready for publication as it is.

We thank the Reviewers for taking the time to read and evaluate our manuscript. We really appreciate their suggestions which substantially improved the manuscript. We have addressed their comments below and revised the manuscript accordingly.

Reviewers' comments:

Reviewer #1 (Remarks to the Author):

Spallacci et al. report a laccase developed using a bioinformatic approach. Overall, this is an interesting piece of work that is of interest so ultimately I would support its publication. However, there are some issues that I'd like to get addressed before publication.

1. The website developed is a bit of a black box. The paper does not specify what exactly how exactly the databases are searched, how many sequences are analyzed, how redundancies are removed, etc. This is important for evaluating this work

Response *We thank the reviewer for highlighting this shortcoming. We have revised the Material and Methods section to clarify the approach of the MetalSite-Analyser, where we also added a new figure (Fig. 5) with a schematic representation of MeSA pipeline.*

2. Kinetic characterization. I feel U/L is not ideal (or meaningful) units. I appreciate the difficulties in establishing purely mol based parameters such as kcat, KM, etc. but U/L is not useful when comparing this work to that of others.

Response. *We thank the reviewer for the insightful comment and fully agree with her/his observation. As stated in the manuscript, the primary objective of this work is to present the adopted design strategy and the experimental characterization of the resulting metallo-peptide complexes. The preliminary activity data are included solely to demonstrate the feasibility and potential of the proposed approach. A comprehensive kinetic and mechanistic characterization will be addressed in future studies.*

We performed additional activity studies with a different laccase substrate analogue and revisited our original activity experiment with ABTS substrate. Based on these new experiments, we concluded that the well-known laccase substrate analogues may not be suitable for kinetic characterization of small metallo-peptide mimics of laccase. We have included these findings in the revised manuscript under 'Electrochemistry and testing of activity' and 'Discussion'.

3. The speciation is a problem. The authors are fairly upfront about the mixed species formation but the overall conclusion need to explicitly address this issue. If the sequences derived from natural sources are so specific, how come such complex speciation is present?

Response. *The main reason complex speciation exists for our minimal peptide is due to the high degree of conformational flexibility of the minimal peptide sequence, which does not have any secondary structure in solution. This is in contrast to the parent protein, where the active site is encapsulated within the protein matrix and preorganized by a network of non-covalent interactions. A key advantage of the minimalistic approach described in this manuscript is to enable studies and understanding of the impact of metal binding towards peptide oligomerization and folding.*

The authors report a difference in conformation between 2:1 and 1:1 peptide:copper ratios. Does this affect the reactivity of the species? Can you relate the reactivity to the changes in character?

Response. *We thank the reviewer for this thoughtful and important question. Indeed, we expect differences in the reactivity between the 2:1 and 1:1 H4pep:Cu complexes. The distinct coordination environments and structural arrangements likely influence the redox properties and, consequently, the catalytic behavior of the respective species.*

However, under the conditions tested, the 1:1 H4pep:Cu complex shows clear evidence of copper release upon reduction. This lability of the metal center compromises the structural integrity of the complex during electrochemical measurements and precludes a direct and meaningful comparison of electrocatalytic activity between the two stoichiometries. As such, while we expect that changes in coordination and conformation impact reactivity, a detailed mechanistic analysis or correlation is not currently possible based on the available data.

We are actively investigating strategies to stabilize the 1:1 complex under reductive conditions in order to enable a more rigorous comparison of the electrocatalytic properties in future work.

The thesis depends on this being a beta-sheet protein with metal centres on the face, and given the structure of the baseline laccase, I feel like it begs the question of what precisely the requirements for kinetic activity are.

Response *We thank the reviewer for raising this insightful and fundamental question regarding the structure–function relationship in catalysis, particularly in relation to catalyst activity and specificity. This issue lies at the core of rational catalyst design and is especially relevant in biomimetic systems where structural simplification is often pursued at the expense of natural complexity.*

We fully agree that the present study only begins to address this broader question. Our manuscript represents a modest but deliberate contribution to this field, using a minimalistic design approach guided by bioinformatics and focused on β -sheet-forming peptide fragments that coordinate copper ions on an accessible face. The MeSA tool we developed is central to this approach, enabling the identification and rational design of short metallopeptides with predefined structural motifs.

By studying simplified systems such as the H4pep:Cu complex, we aim to decouple the complexity of natural metalloenzymes and dissect the individual contributions of secondary structure, metal coordination geometry, and sequence context to catalytic function. While the current system exhibits only modest activity compared to natural laccases, its well-defined and minimal architecture makes it a useful platform for probing structure-function relationships in a controlled and modular way. We addressed these points in the ‘Discussion’ with a more detailed overview, both in structure and function, of the literature examples regarding mononuclear copper-peptide complexes.

Ultimately, insights gained from such simplified models can complement those obtained from larger, more complex systems and may help delineate the essential structural features required for catalysis.

4. A question I have always had relates to the specificity of these types of catalysts and whether it is specific to these sequences or simply due to the promiscuity of coordinated copper- the authors mention similarity to Pecoraro’s coiled coil nitrite reductases. Do these compounds specifically carry out laccase activity (not requiring hydrogen peroxide) or are they promiscuous? It would be interesting to see the reaction scope with different classic laccase

characterizations. There are quite a few classic laccase substrates, of which ABTS is only one. <https://www.mdpi.com/2309-608X/7/2/143> is an excellent resource, and perhaps the authors might consider another common, inexpensive substrate like p-phenylenediamine to determine whether this is a laccase mimic, or simply another ABTS catalyst. Alternatively, they might study the 2,4-dichlorophenol/4-aminoantipyrine reaction that has been reported by several groups reporting laccase mimics. Gazit and co-workers recently reported a Cu(Phe)₂ laccase species with excellent catalytic parameters, and so the authors might wish to compare their protein-based result.

Response. We thank the reviewer for raising this insightful and fundamental question regarding substrate specificity. We performed additional activity studies with a different laccase substrate analogue and revisited our original activity experiment with ABTS substrate. Based on these new experiments, we concluded that the well-known laccase substrate analogues may not be suitable for kinetic characterization of small metallo-peptide mimics of laccase. Our current findings suggest a strong electrostatic interaction between our metallo-peptide mimics and substrate analogues, affecting observable reaction kinetics. We have included these findings in the revised manuscript under 'Electrochemistry and testing of activity' and 'Discussion'.

5. Do the authors observe turnover, in other words is there true catalysis?

Response. Our present preliminary activity studies based on commonly used laccase substrate analogues are insufficient to prove true catalysis. Based on the electrochemical tests conducted, we observed enhanced current that is typical of catalysis under O₂ atmosphere and we are also able to prove that enhanced current is due to O₂ consumption. These are strong indications of electrocatalytic O₂ reduction. However, our preliminary activity investigation, at present, precludes any meaningful determination of catalytic parameters such as turnover numbers or k_{cat} .

6. the authors report the final design as an amalgam of the two fragments they report from Chain A and Chain B of their analysis, however, several elements are confusing to me-The first histidine seems to have been added to mimic the laccase metal site, the second threonine residue comes from selecting the next most conserved residue which is not proline, but I do question the design choices which led to position 4. As far as I can see from the figures, tyrosine is not conserved at all in this position. Did the authors try the different substitutions such as leucine, tryptophan, methionine, and others and not report them? How much "directed evolution" knowledge can we derive from these suggested sequences? It would be interesting to see how general this method is, as it is difficult to rationalize this as a metal binding site predictor should the system require a modification which is not present in the output. A true test of this design would be to use only residues predicted in the alignment.

Response. Tyrosine was included in position 4 to act as a convenient redox-active probe in electrochemical studies. Computational modelling shows that Tyr4 points away from the metal binding sites formed by His residues and thus does not interfere with metal coordination. The main conclusion derived from the MeSA sequence analysis is the HXH motif, present in all four fragments. Since the active site consists of four unconnected fragments and our target is a single minimal peptide sequence, we decided to further simplify the output of MeSA with additional modifications.

The reviewer asks an interesting question on how much ‘directed evolution’ knowledge can be derived from MeSA sequences and how general the predictive power of MeSA approach is. In principle, the MeSA approach should encapsulate salient features relating to the structure of the active site (and perhaps activity as well). In this present study, we did not fully explore the predictive power of MeSA beyond showing, as proof-of-concept, that a molecular Cu metallo-peptide complex with β -sheet motif and positive catalytic O₂ reduction activity can be formed using a minimal peptide sequence derived from the parent protein. A full answer to the reviewer’s question will likely require large combinatorial libraries designed based on MeSA sequences. We have included additional discussions based on the reviewer’s comment in the manuscript.

Minor issues:

Statistical parameters are not always given

Response. *Added statistical parameters, whenever available.*

The “European” way of using comma as opposed to period in e.g. 10.0 is found in many places

Response. *We have corrected the manuscript to use period instead of comma.*

Reviewer #2 (Remarks to the Author):

Major comments:

- 1) Designing is new tool MeSA to design peptides based on minimal functional site of a protein is exciting, however the reviewer could not use the stated link

<https://metalsite-analyzer.cerm.unifi.it/>

Response. *Many thanks to the reviewer for pointing this out. After submission of this manuscript, we noticed that the web server sporadically crashed and needed to be restarted manually. We have identified the cause of the problem which, in short, was due to the data pipeline not producing some of the files required to build the webpages. We have implemented a control routine that checks all the needed tools are properly running or otherwise restarts and recalculates the missing files. If this attempt fails three times, an error message will be provided.*

2) Line 162: “pH 5.6 to prevent copper oxide formation and N-terminal amine deprotonation (vide infra), still favoring imidazole deprotonation and binding (pK_a ~6).” At pH 5.6 most His would be protonated and not be a good ligand for metal. Also pK_a of His would depend a lot on peptide sequence. It is easy to test how much metal ion is binding by running simple ITC experiment. Also CD experiments were performed at pH 4.4, 4.8, 5.2, and 5.6, not much binding of copper would be expected at low pH. Binding of terminal amine to Cu(II) is a valid concern, so protecting NH₂ with acetic anhydride would help to eliminate the binding.

Response. *We thank the reviewer for giving us the opportunity to clarify this point. ITC experiments are generally not conclusive when speciation is directly dependent on cupric ion concentration. We were anyway able to evaluate copper binding apparent dissociation constant at pH 5.6 (Fig. S1). Our results were compared with previously published copper*

binding peptides which were also assayed at similar acidic pH values and reported comparable pK_D for histidine binding (see references 29 and 30 in the main text).

3) For NMR experiment with Cu(I) how the samples were kept under inert atmosphere? Were they dissolved in a degassed buffer? How much dithionite was used? Was the sample in the NMR tube with a screw cap to prevent oxygen from getting in. It is important to see the details as oxidation of copper is possible and then copper could bind to dithionite and Cu(II) would result in broadening of NMR spectra.

Response 3. *According to reviewer's suggestion, we clarified the procedure for Cu(I) experiments under inert atmosphere in the materials and methods section. We have added the following sentence in 'Material and Methods', under 'Spectroscopic methods: UV-vis, CD, EPR, NMR':*

"For Cu⁺ experiments, samples were prepared with deoxygenated buffer and kept under inert Ar atmosphere. Sodium dithionite (3 mM final concentration) was added to maintain reducing conditions. We have not observed any significant difference in NMR spectrum before and after addition of excess of dithionite, therefore excluding dithionite binding or copper oxidation during spectra acquisition."

Moreover, the Cu(II) spectra we acquired are significantly different in terms of line broadening both in CD₃OH and water, with much weaker and unassignable peaks due to paramagnetic shift and broadening.

4) Copper (II) in acetate buffer of 10 mM concentration would be complexed with acetate?

Response. *We thank the reviewer for raising this point. Indeed, in a 10 mM acetate buffer, Cu(II) is expected to form complexes with acetate to some extent. However, our results demonstrate that H4pep binds Cu²⁺ more strongly than acetate under the same conditions. This is supported by EPR titration data (Fig. S10), which clearly shows that all Cu²⁺ ions are coordinated by H4pep, with no indication of free or acetate-bound Cu²⁺ species.*

These findings highlight the high affinity of the peptide for Cu²⁺ and confirm its capacity to outcompete acetate in metal coordination.

Minor:

1) Line 93: Our data demonstrate 92 that, despite its beta-sheet conformation, Cu²⁺(H4pep)₂ does not undergo aggregation, such that we 93 did not detect the formation of fibrils. No aggregation is based on what experiments?

Response. *We have performed dynamic light scattering (DLS) experiments to confirm the absence of aggregates (see figure S12).*

2) Figure 2A: different color spectra correspond to how many equivalent of copper. Band at 786 nm was assigned to free Cu(II), why there is an increased in absorbance when amount of Cu(II) is low (presumably most ions will be bound to peptide)...

Response. *We thank the reviewer for this valuable observation and apologize for not indicating the equivalence of Cu(II) added to the peptide solution in the original submission. We revised*

Figure 2A to indicate the increasing equivalent of Cu(II) added to the peptide solution accordingly to include this information.

The band at 786 nm has been assigned to free, unbound Cu(II). We agree that the increase in absorbance at this wavelength at low Cu(II) equivalents may appear counterintuitive, as most Cu(II) ions are expected to be bound to the peptide under these conditions. The observed absorption band between 500-800 nm region is composed of multiple absorption band. This is indicated by the shifting peak maxima with increasing Cu(II) equivalence. We have fitted three distinct absorption bands in this region of the spectra, with peaks at 636 and 742 nm (assigned to a Cu²⁺-H4pep species) and a peak centered at 786 nm, assigned to free unbound Cu(II). We have amended the manuscript to reflect this clearly and refer readers to Fig. S11 for details of the peak deconvolution procedure.

3) Fig. S1. How much copper was used, buffer pH. These values should be present in figure caption.

Response. We have added the following sentence in the figure S1 caption (Supporting Information)

“...by adding small aliquots of the BCA stock to a solution containing H4pep (40 mM) Cu⁺ complex (prepared with 0.95 equivalents of [Cu(CH₃CN)₄]PF₆ dissolved in acetonitrile up to 10 mM concentration) in 50 mM sodium phosphate buffer at pH 5.8 with 1 mM sodium ascorbate under Argon atmosphere...”

4) In Figure 1B, what is 1 and 2? It is in the main text, but not figure caption.

Response. We have clarified this by adding the following sentence in the Figure 1 caption, section B

“(B) Output of MeSA based on the trinuclear copper site of the model small laccase. 1 and 2 represent fragments belonging to chain A (top) and to chain B (bottom).”

5) No glycerol was used to prepare EPR samples?

Response. We thank the Reviewer for this pertinent question. No glycerol was used in the preparation of the EPR samples. This choice was deliberate, as glycerol at high concentrations (commonly around 30%) can act as a competitive ligand for metal ions such as Cu(II), potentially interfering with the coordination environment of the peptide and altering the spectroscopic features of interest. Moreover, the presence of glycerol would modify the solution composition significantly, introducing variables that could complicate data interpretation.

Our aim was to maintain experimental conditions as consistent as possible with those used in the CD titration experiments, where glycerol was likewise excluded. This approach ensures that any observed spectroscopic changes can be attributed directly to the interaction between the peptide and Cu(II), rather than to changes in solvent environment or competing ligand effects.

6) How oxidation state of copper is controlled in CD experiments?

Response. We thank the reviewer for the opportunity to clarify this question. In the CD experiments, the oxidation state of copper was controlled by using Cu²⁺ salts as the starting material. Cu²⁺ is stable in aqueous solution under aerobic conditions, particularly at the

working pH of 5.6 used in our experiments. No additional redox reagents were introduced, and all manipulations were performed under ambient atmospheric conditions, which further preserves the +2 oxidation state of copper.

Given the absence of reducing agents and the mildly acidic pH, we do not expect any significant reduction of Cu^{2+} to Cu^+ during the course of the experiments. This approach ensures that the observed CD signals reflect the coordination environment of Cu^{2+} with the peptide, without complications from changes in oxidation state.

7) Fig. S10: Stoichiometry – I is missing

Response. *We thank the reviewer for spotting this mistake. We have corrected in Figure S10 caption and also another instance of this mistake in the main text.*

Reviewer #3 (Remarks to the Author):

This manuscript presents a nice study on the design of a minimal biomimetic peptide that mimics the trinuclear copper site of laccase using a bioinformatics-driven approach. The authors describe a novel computational tool, MetalSite-Analyzer (MeSA), to extract metal-binding motifs from metalloenzymes, leading to the rational design of an eight-residue peptide (H4pep) that binds Cu^{2+} and demonstrates catalytic activity. The work is well-motivated and methodologically sound, contributing to the field of bioinspired catalyst design.

However, the design is based on a 4 peptides:3 Cu active site, yet the stoichiometry observed and the model proposed in Fig S16 are quite different. In that sense, the design is not successful. What is the proposed mechanism for the observed activity?

Response. *We agree with the reviewer's comment that the Cu-peptide complexes formed from H4pep do not fully replicate the 4 peptide:3 Cu active site of the parent laccase. Nonetheless, our current investigation shows that an 8-residue minimal peptide sequence, with high degree of conformational flexibility, is able to bind Cu^{2+} via histidine residues to form a molecular complex with β -sheet motif. Moreover, this Cu-peptide complex show positive catalytic activity towards O_2 reduction. We consider this a partial success. We have revised the discussion in the manuscript to highlight this achievement and discuss the utility of these minimal metallo-peptide complexes for structure-function studies.*

We thank the review's suggestion regarding the proposed mechanism. We have included discussions on plausible reaction mechanisms for four-electron reduction of O_2 based on a mono-nuclear Cu complex.

• Did the authors explore alternative sequences with different residues in variable positions? This would help establish whether H4pep is an optimized solution or simply one possibility.

Response. *The reviewer raised an interesting question. This comment is also connected to the comment from reviewer #1, regarding the predictive power of the MeSA approach. In this study we did not explore alternative sequences, therefore we can only state that H4pep is not an optimized solution but rather one of the possibilities. We are in the process of designing new sequences based on MeSA, with more restricted conformation for further structure-activity*

studies. Given the utility of the design approach, we wish to communicate this to the field as soon as possible using H4pep as a proof-of-concept example.

- The EPR analysis, while demonstrating two distinct Cu²⁺-peptide species (1Cu2Pep and 2Cu2Pep), would benefit from a deeper discussion on why only a single axial signal is observed despite presumed symmetry distortions.

Response. *We thank the reviewer for this excellent suggestion. We have included 'Discussion' on the two symmetrically equivalent sites of 2Cu2Pep and how they relate to the 1Cu2Pep site based on additional computational modelling of these two structures.*

- The CD data indicate β -sheet formation upon Cu²⁺ binding, but no direct structural characterization (e.g., X-ray crystallography or NMR-derived structure) is provided. Can molecular dynamics simulations or secondary-structure predictions further support this conclusion?

Response: *We thank the reviewer for this insightful suggestion. As correctly noted, the CD data indicate β -sheet formation upon Cu²⁺ binding, but no direct structural data (e.g., X-ray or NMR) are currently available for these complexes. While we acknowledge the limitations in the current experimental structural characterization, we agree that computational approaches could offer additional support.*

Regarding molecular dynamics (MD) simulations, we note that this approach is not ideally suited for modeling metal–ligand interactions in this context, as classical force fields do not adequately describe the bond formation and breaking events that are often critical in metal coordination chemistry. Therefore, (MD) simulations present several challenges, as the simulation of metalloproteins is highly dependent on the parametrization of metal centers in the chosen force field, and depending on the specific parametrization, the coordination geometry is sometimes not preserved.

We have revised our manuscript to include geometry optimization of the 1Cu2Pep and 2Cu2Pep complexes to provide plausible structural models. However, we emphasize that such models represent only one of the many possible low-energy conformations and should be interpreted with caution. Due to the inherent flexibility of the peptide and the complexity of metal coordination, multiple conformers are likely accessible under the experimental conditions.

DSL measurements at high concentration of peptide show no aggregates, but are also nonreproducible. Have the authors tried sedimentation ultracentrifugation or SEC? How about AFT or TEM, which might capture structures relevant to the observed electrochemical activity?

Response. *We thank the reviewer for this thoughtful and constructive comment. As noted, DLS measurements were performed at high peptide concentrations and did not show aggregates. However, the data were not reproducible, which is consistent with the limitations of the technique in the absence of particulate species. DLS is inherently more reliable when aggregates/particles are present, and its sensitivity and reproducibility decrease significantly when the sample is largely monodispersed or consists of small, non-scattering species.*

At this stage, we have not yet performed sedimentation velocity analytical ultracentrifugation (AUC) or size-exclusion chromatography (SEC), although we acknowledge that these could

provide complementary insights. These methods will be considered for future studies to better resolve the solution behavior of the complexes.

With regard to microscopy-based techniques such as AFM or TEM, while they are powerful for visualizing nanostructures, they do not offer molecular-level speciation. In our case, they would not reliably distinguish between relevant Cu-peptide aggregates and extraneous particles (e.g., electrode polishing debris or dust), which could lead to ambiguous interpretations. Therefore, we have opted not to employ these techniques at this stage.

- The electrocatalytic activity of 1Cu2Pep is intriguing but remains modest in comparison to natural laccases.

Response. *We appreciate the reviewer's comment and fully agree that the electrocatalytic activity observed for the 1Cu2Pep complex remains modest in comparison to that of natural laccases. This is indeed expected, given the remarkable complexity and evolutionary optimization of natural multicopper oxidases, which feature highly organized active sites, multiple redox centers, and finely tuned proton and electron transfer pathways.*

Our objective with this study was not to replicate the full catalytic efficiency of laccases, but rather to demonstrate that minimal peptide fragments, derived from bioinformatically guided sequence selection, can support copper coordination and exhibit measurable redox activity. In this context, the observed activity serves as a proof-of-concept, showing that even structurally simple systems can recapitulate some functional features of natural enzymes.

We will highlight this point in the revised manuscript to set realistic expectations and to highlight the significance of our approach as a modular and tunable platform for the future development of more efficient artificial metalloenzymes.

- The faradaic efficiency (62%) suggests that side reactions (possibly O₂ reduction to H₂O₂) may be occurring. Have the authors tested for peroxide formation?

Response. *We have reanalyzed our electrochemistry data and realized there was a mistake when performing O₂ calibration of our custom Clark type electrode. We have revised the analysis based on this new calibration and taken into account the reviewer's comment on possible two-electron reduction to H₂O₂. We have revised our manuscript and the present data do not distinguish between two or four-electron reduction of O₂. As such we have not stated a Faradaic efficiency parameter in the revised manuscript but presented alternative scenarios. In our revised analysis, we take advantage of our simultaneous O₂ concentration and coulometry data and present two scenarios (i) four-electron reduction of O₂ where faradaic efficiency approaches unity and (ii) two-electron reduction of O₂ with halved faradaic efficiency.*

In addition, we also included a discussion on plausible reaction mechanisms for four-electron reduction of O₂ with a mononuclear Cu complex.

- At -0.30 V, the increasing current suggests an alternate active species forming over time. Can spectroscopic or electrochemical evidence confirm this?

Response. *At present, we are not able to detect any alternative active species formed over time during electrolysis. The main purpose of the present manuscript is to report on the proposed methodology to design minimal metallo-peptide complexes as simplified models for in-depth structure-function relationship studies. The reported catalytic activity serves as proof-of-*

concept of this approach. The intriguing activity and redox behavior observed in our preliminary activity studies reported here will motivate us (and hopefully others) to elucidate this in a more in-depth study on the redox and catalytic properties of these metallo-peptide complexes in follow-up studies.

- The cyclic voltammetry rinse tests suggest no metal deposition for 1Cu2Pep, but what about long-term stability? Have the authors examined whether repeated catalysis deactivates the peptide?

Response. *We thank the review for this suggestion. In this manuscript, we did not carry out long term stability test for **1Cu2Pep** as the intent was only to show positive activity as proof-of-concept for our design approach. We defer critical evaluation of catalytic performance, including kinetic, mechanistic and stability investigations to a separate study. We note that there are different norms for carrying out long-term stability tests in catalysis, where reported duration for stability studies ranges from minutes to days.*

*In the present manuscript, the rinse test was performed to exclude metal deposition from **1Cu2Pep** in order to evaluate whether O₂ reduction activity originates from **1Cu2Pep** or from deposited Cu on electrode.*

- The study would benefit from a comparison with other reported biomimetic metal-peptide systems. For instance:

- How does the catalytic efficiency of 1Cu2Pep compare with previously reported Cu²⁺-peptide complexes?
- Could a comparison to engineered metalloproteins or synthetic small-molecule catalysts provide better context for H4pep's catalytic performance?

Response: *We thank the reviewer for this important suggestion. We agree that including a comparison with other reported biomimetic Cu²⁺-peptide systems, engineered metalloproteins, and synthetic small-molecule catalysts would provide useful context for evaluating the catalytic performance of the **1Cu2Pep** complex. In the present manuscript, we did not set out to derive quantitative parameters of catalytic performance which makes comparison to literature difficult. Also, in our revised manuscript, we note that the observed activity based on standard laccase substrates appears to be highly influenced by electrostatic interaction, hinting towards incompatibility of these assays for activity determination of small metallo-peptide mimics. We have included these discussions in the revised manuscript.*

- The figures in the main text and supplement are generally clear, but some could be improved:
- Figure 2D (speciation diagram) would benefit from an explanation of how relative species distributions were derived.
- Supplementary Figure S11 (UV-Vis deconvolution) should include additional discussion on fitting assumptions.
- EPR simulations (Fig. S9–S10) should be described in more detail for readers unfamiliar with spectral deconvolution.

Response: *We thank the reviewer for pointing this out, thus giving us the opportunity to clarify the steps used for UV-Vis deconvolution and EPR simulations.*

We have added a paragraph in the Supplementary Materials, describing spectral deconvolution and highlighting these aspects.

- The introduction is well-written but somewhat lengthy. Consider streamlining background information and moving some discussion on metalloenzyme active sites to the discussion section. This is especially in view of the fact that the design of a mini-laccase is not successful.
- The discussion sometimes reiterates results rather than providing deeper insights. A more concise, focused discussion on broader implications (e.g., potential for enzyme-inspired catalysis in industry) would improve readability.

Response. *We thank the reviewer for this critical comment. We have thoroughly revised the 'Discussion' section based on the reviewer's suggestions and revised the 'Introduction' section accordingly to emphasize on the design approach. We believe the manuscript is significantly improved after this revision.*

- Minor grammatical errors are present. For example:
 - “A promising strategy for designing biomimetic catalysts holds on mimicking...” → “A promising strategy for designing biomimetic catalysts relies on mimicking...”
 - “Nature-inspired or biomimetic catalyst aims to reach”... should be “catalysts aim...”

Response. *We thank the reviewer's suggestions and assistance in improving the readability of the manuscript. We have revised the manuscript accordingly.*

We consider the Reviewers' contribution of great value as it considerably helped us in improving the final version of our manuscript.

Reviewers' comments:

Reviewer #1 (Remarks to the Author):

The manuscript is improved enough to warrant publication in the present form.

We are grateful that the manuscript is now suitable for publication according to the Reviewer's opinion. His/her suggestions have been crucial to achieve the final result.

Reviewer #2 (Remarks to the Author):

Most of the questions were addressed, however there remain several points that need further explanation.

We apologize that some points are still needing a clearer explanation. We hope that we could now address the remaining concerns that have been raised by the Reviewer.

1. The authors look at binding of Cu(I) to BCA, however Cu(II) is binding to peptides and not Cu(I), so I am confused by why binding affinity for Cu(I) was measured.

We performed Cu(I) titration by BCA to demonstrate that the copper is stably bound by the H4pep in both oxidation states (as shown by NMR, UV/Vis, and EPR spectroscopies). This is particularly relevant considering that the peptide is able to activate dioxygen-dependent oxidation catalysis. Moreover, we performed this competitive titration to compare the binding properties of our complex to the literature values. By evaluating K_D constant, we were able to assess that peptide affinity for copper(I) is in the same range of previously published His-based peptide ligands. The determination of Cu(II) affinity for H4pep was unfortunately too complex because of the formation of two species with different nuclearity (1Cu2Pep and 2Cu2Pep).

2. For the NMR experiment, Zn(II) is used as a model for paramagnetic Cu(II) and conclusions are made for which ligands bind to the metal ion. But Zn(II) is not a great substitute for Cu(II) as it prefers different ligands and coordination geometry (Cu(II) is octahedral and Zn(II) is tetrahedral, Zn(II) has more preference for S compared to Cu(II), which prefers N ligands). Did anyone report use of Zn(II) as a substitute for Cu(II)?

We thank the reviewer for raising this important point. As a d^{10} ion, Zn^{2+} does not show significant preference towards a specific geometry and follows copper in the Irving-Williams series of divalent metal cations. The use of Zn(II) as a diamagnetic probe in NMR studies is a strategy used to gain preliminary structural insights when the paramagnetic nature of Cu(II) precludes direct NMR analysis [1,2]. In such cases, Zn(II) allows for the identification of potential binding sites and ligand interactions without the complications associated with paramagnetic broadening. Zn(II) complexation in tetrahedral geometries can be found in the PDB in mixed sulphur/nitrogen ligand spheres (zinc fingers, zinc-substituted cyanins and cupredoxins) as well as in nitrogen ligand spheres (carbonic anhydrase, Cu/Zn SOD). Most interestingly, zinc substitution in natural LPMOs does not significantly alter the orientation of the first coordination shell also in its particularly demanding pentacoordinated geometry [1,3].

In our study, we also performed NMR analysis (titration) with Cu(I) (at H4pep:Cu⁺ ratios of 1:1 and 2:1) under anaerobic conditions both in aqueous buffer (Fig. S2C and S5) and in methanol (Fig. S4B and S6). Similar to the Zn(II) complexes, we observed downfield shifts of histidine resonances and pronounced signal broadening,

suggesting that the peptide behaves similarly with both cations. This indicates that the Zn(II) substitution has a minimal effect on the overall coordination mode, at least in the case of the d^{10} Cu(I) cation.

Zn(II) was used strictly for this purpose — to probe ligand coordination via NMR in a system where Cu(II) itself would render complicate spectral interpretation.

We fully agree that conclusions based on Zn(II) coordination should be interpreted with caution. We are aware that if H4pep binds Cu(II) in a square planar (Jahn-Teller axially distorted octahedral) coordination, such geometry cannot be easily obtained with zinc substitution, nonetheless these experiments gave us crucial information about the overall oligomerization/stoichiometry, as well as clear indication of which amino acids are involved in metal binding.

1. F.L. Aachmann, M. Sørlie, G. Skjåk-Bræk, V.G.H. Eijsink, G. Vaaje-Kolstad, NMR structure of a lytic polysaccharide monooxygenase provides insight into copper binding, protein dynamics, and substrate interactions. *Proc Natl Acad Sci USA*, **109**, 18779–18784, (2012).
2. L. Hou, M. G. Zagorski, NMR Reveals Anomalous Copper(II) Binding to the Amyloid Ab Peptide of Alzheimer's Disease. *J. Am. Chem. Soc.*, **128**, 9260-9261, (2006).
3. Z. Forsberg, A.K. Mackenzie, M. Sørlie, Å.K. Røhr, R. Helland, A.S. Arvai, G. Vaaje-Kolstad, & V.G.H. Eijsink, Structural and functional characterization of a conserved pair of bacterial cellulose-oxidizing lytic polysaccharide monooxygenases, *Proc. Natl. Acad. Sci. USA*, **111**, 8446-8451, (2014).

REVIEWERS' COMMENTS:

Reviewer #2 (Remarks to the Author):

I reviewed the responses to my earlier comments and this that the paper is ready for publication as it is.

We are grateful that the manuscript is now suitable for publication according to the Reviewer's opinion. The reviewers comments and suggestions has improved the manuscript.